



# Investigation of the limonene photooxidation by OH at different NO concentrations in the atmospheric simulation chamber SAPHIR

Jacky Y. S. Pang[1], Anna Novelli[1], Martin Kaminski[1,a], Ismail-Hakki Acir[1,b], Birger Bohn[1], Philip T. M. Carlsson[1], Changmin Cho[1,c], Hans-Peter Dorn[1], Andreas Hofzumahaus[1], Xin Li[1,d], Anna Lutz[2], Sascha Nehr[1,e], David Reimer[1], Franz Rohrer[1], Ralf Tillmann[1], Robert Wegener[1], Astrid Kiendler-Scharr[1], Andreas Wahner[1], and Hendrik Fuchs[1]

[1]Institute of Energy and Climate Research, IEK-8: Troposphere, Forschungszentrum Jülich GmbH, Jülich, Germany

[2]Department of Chemistry and Molecular Biology, University of Gothenburg, Gothenburg, Sweden

[a]now at: Federal Office of Consumer Protection and Food Safety, Department 5: Method Standardisation, Reference Laboratories, Resistance to Antibiotics, Berlin, Germany

[b]now at: Institute of Nutrition and Food Sciences, Food Science, University of Bonn, Bonn, Germany

[c]now at: Atmospheric Trace Molecule Sensing Laboratory, School of Earth Sciences and Environmental Engineering, Gwangju Institute of Science and Technology, Gwangju, Republic of Korea

[d]now at: State Key Joint Laboratory of Environmental Simulation and Pollution Control, College of Environmental Sciences and Engineering, Peking University, Beijing, China

[e]now at: CBS International Business School, Brühl, Germany

*Correspondence to*: Hendrik Fuchs (h.fuchs@fz-juelich.de)

**Abstract.** The oxidation of limonene by the hydroxyl (OH) radical and ozone ($O_3$) was investigated in the atmospheric simulation chamber SAPHIR in experiments performed at different nitric oxide (NO) mixing ratios from nearly zero up to 10 ppbv. For the experiments dominated by OH oxidation the formaldehyde (HCHO) yield was experimentally determined and found to be ($12\pm3$), ($13\pm3$), and ($32\pm5$) % for experiments with low (~0.1 ppbv), medium (~0.3 ppbv), and high NO (5 to 10 ppbv), respectively. The yield in an ozonolysis-only experiment was ($10\pm1$)%, which agrees with previous laboratory studies. The experimental yield of the first generation organic nitrates from limonene-OH oxidation is calculated as ($34\pm5$)%, about 11% higher than the value in the Master Chemical Mechanism (MCM), which is derived from structure-activity-relationships (SAR). Time series of measured radicals, trace-gas concentrations, and OH reactivity are compared to results from zero-dimensional chemical box model calculations applying the MCM v3.3.1. Modelled OH reactivity is 5 to 10 s$^{-1}$ (25% to 33% of the OH reactivity at the start of the experiment) higher than measured values at the end of the experiments at all chemical conditions investigated, suggesting either that there are unaccounted loss processes of limonene oxidation products or that products are less reactive toward OH. In addition, model calculations underestimate measured hydroperoxyl radical ($HO_2$) concentrations by 20% to 90% and overestimate organic peroxyl radical ($RO_2$) concentrations by 50% to 300%. Largest deviations are found in low-NO experiments and in the ozonolysis experiment. An OH radical budget analysis, which uses only measured quantities, shows that the budget is closed in most of the experiments. A similar budget analysis for $RO_2$ radicals



suggests that an additional $RO_2$ loss rate of about $(1-6) \times 10^{-2}$ s$^{-1}$ for first-generation $RO_2$ is required to match the measured $RO_2$ concentrations in all experiments. Sensitivity model runs indicate that additional reactions converting $RO_2$ to $HO_2$ at a rate of about $(1.7-3.0) \times 10^{-2}$ s$^{-1}$ would improve the model-measurement agreement of $NO_x$, $HO_2$, $RO_2$ concentrations, and OH reactivity. Reaction pathways that could lead to the production of additional OH and $HO_2$ are discussed, which include isomerisation reactions of $RO_2$ from the oxidation of limonene, different branching ratios for the reaction of $RO_2$ with $HO_2$, and a faster rate constant for $RO_2$ recombination reactions. As the exact chemical mechanisms of the additional $HO_2$ and OH sources could not be identified, further work needs to focus on quantifying organic product species and organic peroxy radicals from limonene oxidation.

## 1 Introduction

About 1000 Tg of biogenic volatile organic compounds (BVOCs) are emitted into the atmosphere every year. Approximately half of the emissions comprise isoprene while 15% consist of monoterpene species (Guenther et al., 2012). Among monoterpenes, limonene is the fourth most abundant species, which comprises about 10% of the total monoterpene emissions (Sindelarova et al., 2014). Apart from biogenic sources, limonene is also emitted from anthropogenic sources such as air fresheners and cleaning products (Liu et al., 2004; Nazaroff and Weschler, 2004; McDonald et al., 2018; Gkatzelis et al., 2021a, b). Limonene has two carbon-carbon double bonds, which make it reactive towards atmospheric oxidants such as ozone ($O_3$), hydroxyl radicals (OH), and nitrate radicals ($NO_3$) (Calogirou et al., 1999; Atkinson and Arey, 2003).

During daytime, the most important oxidant is the hydroxyl radical (OH). The reaction between the olefin group and the OH radical yields a β-hydroxyalkyl radical, which subsequently reacts with an oxygen molecule ($O_2$) under atmospheric conditions to form an organic peroxy radical ($RO_2$) (Fig. 1). In the presence of nitric oxide (NO), $RO_2$ is converted to an alkoxy radical (RO) and thereby NO is oxidized to nitrogen dioxide ($NO_2$). $NO_2$ can be photolyzed back to NO together with an oxygen atom, which then reacts with an oxygen molecule producing ozone. This mechanism is the most relevant source for tropospheric ozone. Alkoxy radicals are very reactive and they often quickly decompose forming carbonyl products and a hydroperoxyl radical ($HO_2$) or further organic radicals (Orlando et al., 2003). OH is then regenerated through the reaction between $HO_2$ and NO, closing the radical reaction cycle and producing another ozone molecule from the photolysis of nitrogen dioxide. In polluted environments with high $NO_x$ (=NO+$NO_2$) emissions such as in urban areas (NO concentration > 1000 pptv; e.g., Dusanter et al. (2009) and Lu et al. (2017)), the reaction between $RO_2$ and NO is often the dominant pathway, through which OH is regenerated. In remote regions with low NO mixing ratios (< 200 pptv; e.g., Ren et al. (2006) and Whalley et al. (2011)), $RO_2$ predominantly reacts with $HO_2$ and forms an organic peroxide molecule. This process terminates the radical chain. The organic peroxide produced during this process can potentially add to secondary organic aerosols (SOA) (Surratt et al., 2006). $RO_2$ radicals can also react with other $RO_2$ radicals. The reaction rate constant is usually low ($k$ ranges from $10^{-15}$ to $10^{-11}$ cm$^{-3}$ s$^{-1}$; (Tyndall et al. (2001); Jenkin et al. (2019)) and the $RO_2$ loss rate is often small for atmospheric $RO_2$ concentrations ($RO_2$



~ $10^8$ cm$^{-3}$; Tan et al. (2018)). Products of the RO$_2$ recombination reactions are either alkoxy radicals or oxidized organic compounds containing alcohol or carbonyl functional groups, which terminate the radical reaction chain.

During the last decade, there has been increasing evidence from laboratory studies as well as theoretical studies that also unimolecular reactions of RO$_2$ can be of importance in the atmosphere. Unimolecular H-shift reactions of RO$_2$ can promote the formation of low volatility organic compounds as the number of oxygen atoms in the molecule quickly increases during the process (Jokinen et al., 2014). In some of the unimolecular reactions, HO$_x$ (=OH+HO$_2$) radicals can be regenerated, so that this reaction pathway becomes a radical propagation reaction that does not require the presence of NO. This has been, for example, shown for isoprene (Peeters and Müller, 2010; Crounse et al., 2011; Peeters et al., 2014; Berndt et al., 2019; Novelli et al., 2020) and methacrolein (Crounse et al. 2012, Fuchs et al., 2014). The reaction rate constant of H-shift unimolecular reactions depends on the position of the H-atom relative to the peroxy group, the position of other functional groups to the H-atom, and temperature (Crounse et al., 2013; Vereecken and Nozière, 2020). H-shift reactions are very slow ($k < 10^{-3}$ s$^{-1}$ at 298 K) in an aliphatic peroxy radical without a nearby electron drawing functional group for atmospheric conditions (Otkjær et al., 2018; Praske et al., 2019), so that they typically cannot compete with bimolecular reactions ($k_{bi} \sim 10^{-2}$ s$^{-1}$, for 50 pptv of NO and $5 \times 10^8$ cm$^{-3}$ of HO$_2$). Even if the RO$_2$ radical contains a hydroxyl group, rate constants for H-shift reactions from the hydroxyl group are still low ($k \sim 10^{-2}$ to $10^{-1}$ s$^{-1}$) (Vereecken and Nozière, 2020) and can only compete with bimolecular reactions in the atmosphere in remote areas or suburban regions with low to moderate NO concentrations (50 – 500 pptv). However, there are also fast H-shift reaction pathways that could compete with bimolecular reactions even in moderately or heavily polluted scenario such as allylic-H shift ($k \sim 1$ s$^{-1}$) or aldehydic H-shift ($k \sim 1 - 100$ s$^{-1}$) reactions, in which the radical is stabilized by delocalized electrons or the carbonyl group, respectively (Otkjær et al., 2018; Zhang and Dibble, 2011; Vereecken and Nozière, 2020). Concerning limonene, which contains two carbon-carbon double bonds (Fig. 1), recent theoretical studies suggest that the RO$_2$ radical formed after the reaction with OH or O$_3$ undergoes rapid unimolecular reactions (reaction rate constants range from 1 to $10^2$ s$^{-1}$; Møller et al., 2020; Chen et al., 2021), which would be competitive with bimolecular reactions even in polluted environments. In this study, the atmospheric degradation of limonene was investigated in the atmospheric simulation chamber SAPHIR (Simulation of Atmospheric PHotochemistry In a large Reaction Chamber) at Forschungszentrum Jülich, Germany. Experiments were performed at three atmospherically relevant NO concentrations ranging from 0.1 to 10 ppbv. In addition, one ozonolysis experiment was conducted without the presence of NO$_x$. The main objective of this work is to evaluate the performance of the current chemical model (Master Chemical Mechanism version 3.3.1, http://mcm.leeds.ac.uk/MCM) by comparing observations of trace gas and radical concentrations with model results. In particular, the question of whether the regeneration of OH from the radical cycle can be described by model calculations is investigated.



**Figure 1.** Simplified mechanism of the limonene oxidation by OH and O₃. If available, names of the species as they appear in the MCM v3.3.1 are given. For simplicity the stereo-specificity of the intermediates, RO₂ reactions with HO₂ and other RO₂, as well as the formation of organic nitrates are not shown here. Production of HCHO from the first oxidation step of limonene is labelled in red; HCHO production from the second oxidation step is labelled in blue; production of a HO₂ radical is labelled in orange. The production of HCHO from the decomposition of LIMALBO is not classified as it is not produced from the oxidation of the terminal C=C double bond. Most of the mechanism and branching ratios are taken from the MCM v3.3.1, except for the branching ratios of the O₃ addition at the external C=C bond of limonene (Wang and Wang, 2021) and the decomposition of LIMOOB to L5O2.



## 2 Methods

### 2.1 Atmospheric simulation chamber SAPHIR

SAPHIR is a large outdoor chamber located at Forschungszentrum Jülich, Germany. The cylindrical-shaped chamber (18 m length, 5 m diameter, volume 270 m$^3$) is made of a double-wall Teflon (FEP) film, which minimizes wall loss and allows the transmission of the entire spectrum of solar radiation. A shutter system can be opened and closed to allow experiments in illuminated or dark conditions. All experiments are performed in synthetic air produced by evaporating ultrapure liquid oxygen and nitrogen (Linde, purity > 99.9999%). The air pressure inside the SAPHIR chamber is kept above the atmospheric pressure by 35 Pa to ensure that air outside the chamber cannot leak into the chamber, resulting in a typical dilution rate of trace gases of approximately 4 % h$^{-1}$. The temperature in the chamber is similar to the ambient temperature. More details of the chamber can be found in previous publications Rohrer et al. (2005) and Kaminski et al. (2017).

### 2.2 Instrumentation

Table 1 provides an overview of the quantities that were measured and the corresponding instruments. NO and NO$_2$ concentrations were monitored with a chemiluminescence instrument (Eco Physics) with a blue-light photolytic converter that converts NO$_2$ to NO. In some experiments, the zero value of the instrument was not accurately determined as could be seen by a significant NO and NO$_2$ mixing ratio (NO < 0.02 ppbv, NO$_2$ < 0.5 ppbv) that was measured in the dark, clean chamber. For these experiments, the instrumental zero was subtracted for the analysis of data in this work. Nitrous acid was measured in some experiments by a long-path absorption photometer (Li et al., 2014). Similar as for the NO$_x$ instruments, non-zero HONO (< 100 pptv) was measured in the clean dark chamber in some experiments. This value was attributed to an unaccounted instrumental zero value and subtracted from the data.

O$_3$ was measured with a UV absorption instrument (Ansyco). Photolysis frequencies were calculated from solar actinic flux densities measured by a spectroradiometer outside the chamber (Bohn et al., 2005; Bohn and Zilken, 2005).

OH was measured with a laser-induced fluorescence (LIF) instrument, in which OH is excited by laser light at a wavelength of 308 nm in a low-pressure cell (3.5 hPa) (Holland et al., 1995). HO$_2$ was measured by the LIF instrument in a separate detection cell (HO$_x$-cell), in which HO$_2$ is first converted to OH in the reaction with NO and the sum of OH and HO$_2$ (=HO$_x$) is detected by OH fluorescence. The HO$_2$ concentration is then calculated by subtracting the OH concentration measured in the OH-cell from the signal obtained in the HO$_x$-cell. Organic peroxy radicals (RO$_2$) can potentially interfere in the measurement of HO$_2$ due to the concurrent conversion of RO$_2$ to HO$_2$ after the reaction with NO. This interference is usually less than 5% for simple alkoxy radicals (C$_1$-C$_4$) because of the slow conversion rate of the alkoxy radical produced in the reaction of RO$_2$ with NO at the reduced O$_2$ concentration in the low-pressure cell. However, β-hydroxy alkoxy radicals formed from the OH-initiated oxidation of larger alkene species (e.g., isoprene) could rapidly decompose and then react with O$_2$ to





form HO$_2$, which adds to the sampled HO$_2$ radical concentration (Fuchs et al., 2011; Lu et al., 2012). During the experiments in this work, the HO$_2$ cell was operated with low NO concentrations to minimize the conversion of RO$_2$ to HO$_2$ radicals, but it cannot be excluded that a small fraction of limonene RO$_2$ acted as interference.

RO$_2$ was measured in the RO$_x$-LIF system, which consists of a converter mounted on top of a fluorescence detection cell. In

the converter, RO$_2$ is converted to HO$_2$ in the presence of added CO and NO at low total pressure (~ 25 hPa). The HO$_2$ is then passed into a low-pressure detection cell (3.5 hPa) where HO$_2$ is converted into OH by reaction with a large excess of added NO. The RO$_2$ concentration is calculated by subtracting the HO$_x$ concentration from the measured signal (Fuchs et al., 2008, 2011). The sensitivity of the LIF instrument to RO$_2$ is regularly calibrated for methylperoxy radicals. Laboratory tests show that the instrument's sensitivity for RO$_2$ radicals from limonene might be slightly reduced compared to the sensitivity for

methylperoxy radicals (0.85±0.05).

Apart from the detection by the LIF instrument, OH radical concentrations were measured by differential optical absorption spectroscopy (DOAS) (Dorn et al., 1995) in the experiments in 2015. The mean differences between the LIF-measured OH and DOAS-measured OH concentrations in the two experiments in 2015 were -13 % and +39 %. The difference between measurements on the latter experiment was higher than the combined accuracies of the measurements. Because DOAS is

inherently a calibration-free method and it is regarded as a reference method (Schlosser et al., 2007, 2009), OH concentrations from this instrument are used for the evaluation of experiments in 2015.

Measurements of the OH reactivity ($k_{OH}$), the inverse chemical lifetime of the OH radical, were achieved by an instrument making use of laser flash-photolysis combined with the OH detection by LIF (Lou et al., 2010; Fuchs et al., 2017). In this instrument, air is sampled through a flow tube. A high concentration of OH is generated by ozone photolysis in the presence

of water vapour using a short laser pulse at 266 nm, followed by the time-resolved measurement of the OH concentration while the OH is consumed by OH reactants contained in the sampled air. The pseudo-first-order rate loss constant of the decay of the OH concentration gives the OH reactivity value.

Limonene concentrations were measured by gas chromatography coupled with a flame ionization detector (GC-FID, Kaminski (2014)), as well as by a proton-transfer-reaction time-of-flight mass spectrometer (PTR-TOF-MS) (Lindinger et al., 1998;

Jordan et al., 2009). Two GC-FIDs were used in the experiments in 2012. The differences between measurements by the 2 GC-derived measurements for limonene concentrations were about 12%. Discrepancies between the limonene concentrations measured by GC-FID and PTR-MS were about 12% depending on which GC-FID measurement is used.

The amount of injected limonene can also be derived from the rapid increase of OH reactivity, when limonene is injected. The injected limonene concentrations derived from the increase of OH reactivity agree with the PTRMS measurements within 15%

in three experiments (08 August 2012, 10 August 2012, and 13 June 2015), but discrepancies were between 20 – 55% in the other experiments. For the analysis in this work, PTR-MS measurements were scaled in some experiments to match the



limonene injections assuming that the measurement of OH reactivity is more accurate than PTR-MS measurements because no calibration of the instrument is required.

A time-of-flight chemical ionization mass spectrometer (ToF-CIMS) applying ionization with nitrate ions ($^{15}NO_3^-$) detected
some nitrated oxidized species in one limonene-OH oxidation experiment with high NO concentrations (12 August 2013) (Zhao et al., 2018). Although there was no calibration available to derive absolute concentrations, the time series of mass to charge ratios gave information about the relative abundance of these species.

Measurements of formaldehyde (HCHO) concentrations were available from either one of three instruments which included an instrument making use of the Hantzsch reaction (AL4021, Aero Laser GmbH), the DOAS instrument that is also used for
the detection of OH, and a cavity ring-down spectroscopy (CRDS) analyzer (Picarro G2307) (Glowania et al., 2021). Measurements using the Hantzsch method were available in the experiments in 2012; measurements with DOAS were available in the experiments in 2015; measurements with CRDS were available in the experiments in 2019. Measurements were corrected for an unaccounted instrumental zero value (< 0.5 ppbv) observed in the clean dark chamber in some experiments.








**Table 1.** Instrumentation for radical and trace-gas measurements in the chamber experiments.

| Species | Method | Time resolution | 1 σ precision | 1 σ accuracy |
|---|---|---|---|---|
| OH | DOAS[a] | 205 s | $0.8 \times 10^6$ cm$^{-3}$ | 6.5 % |
|  | LIF[b] | 47 s | $0.3 \times 10^6$ cm$^{-3}$ | 13 % |
| HO$_2$, RO$_2$ | LIF | 47 s | $1.5 \times 10^7$ cm$^{-3}$ | 16 % |
| OH reactivity | Laser flash photolysis + LIF | 180 s | 0.3 s$^{-1}$ | 0.5 s$^{-1}$ |
| NO | Chemiluminescence | 60 s | 20 pptv | 5 % |
| NO$_2$ | Chemiluminescence + photolytical converter | 60 s | 20 pptv | 5 % |
| O$_3$ | Chemiluminescence | 180 s | 60 pptv | 5 % |
| Limonene | PTR-TOF-MS[c] | 40 s | 15 pptv | 14 % |
|  | GC-FID[d] | 45 min | 4 – 8 % | 5 % |
|  | DOAS[e] | 100 s | 20 % | 7 % |
| Formaldehyde | CRDS[f] | 300 s | 90 pptv | 10 % |
|  | Hantzsch[g] | 60 s | 25 pptv | 8.6 % |
| Acetone | GC-FID | 45 min | 4 – 8 % | 5 % |
| HONO | LOPAP[h] | 300 s | 3 pptv | 10 % |
| Photolysis frequencies | Spectroradiometer | 60 s | 10 % | 18 % |

[a] Differential optical absorption spectroscopy. [b] Laser-induced fluorescence. [c] Proton transfer reaction time-of-flight mass spectrometry. [d] Gas chromatography coupled with flame ionization detector. [e] Differential optical absorption spectroscopy, available for the experiments in 2015. [f] Cavity ring-down spectroscopy, available for the experiments in 2019. [g] not available in the experiments on 12 August 2013 and 05 June 2020 [h]Long-path absorption photometer.


**2.3 Limonene oxidation experiments**

Before the start of an experiment, the chamber was cleaned by flushing dry, ultra-pure synthetic air through the chamber overnight to purge out trace gases that remained from previous experiments. To humidify the air at the start of the experiment, water vapor from boiling ultra-pure water (Milli-Q) was flushed into the chamber, until the relative humidity reached about

70%. The relative humidity gradually decreased as a result of the increasing temperature in the chamber over the course of the experiment and the dilution by the dry replenishing air. Before the injection of limonene, the chamber roof was opened to allow sunlight to irradiate the clean chamber air (zero air phase). In the illuminated chamber, small amounts of nitrous acid (HONO), formaldehyde, and acetone were formed with a rate of a few hundred pptv h$^{-1}$ (Rohrer et al., 2005). Therefore, the





primary source of OH and NO in most of the experiments was the photolysis of HONO leading also to a continuous increase

of nitrogen oxide concentrations ($NO_x$).

In total, seven experiments investigating limonene oxidation were performed. Chemical conditions can be divided by the NO concentration levels (Table 2), for which the contributions of different $RO_2$ loss reactions varied between radical propagation reactions (i.e., reaction with NO), radical termination channels (i.e., reactions with $HO_2$ and $RO_2$) and isomerisation reactions.

In the experiments with low NO mixing ratios of 0.1 to 0.15 ppbv (01 September 2012 (Fuchs et al., 2021a), 04 July 2019

(Bohn et al., 2021b)), about 3 to 4 ppbv of limonene was injected three times. Between each injection, limonene was oxidized for about 90 to 120 minutes, so that most of the limonene reacted away before the next injection. To suppress NO concentrations during the experiments, approximately 50 to 60 ppbv of $O_3$ produced by a silent discharge ozonizer (O3Onia) was injected before opening the chamber roof.

In experiments with medium NO concentrations ranging from 0.25 to 0.4 ppbv (08 and 10 August 2012; Fuchs et al., 2021a,

b)), about 4 ppbv limonene was injected two hours after opening the roof. No additional trace gases were added. Measured $O_3$ mixing ratios increased from about 1 ppbv to 10 ppbv as a result of the photolysis of $NO_2$, which was produced from the reaction between peroxy radicals and NO.

In the experiment with the high NO concentrations (03 August 2015, (Bohn et al., 2021c)), about 15 ppbv of NO was injected into the chamber before opening the chamber roof and 10 ppbv of limonene at later times. When most of the limonene was

consumed within two hours after the first injection, an additional injection of 10 ppbv of limonene was done.

Lastly, a limonene ozonolysis experiment was conducted on 05 June 2020 (Bohn et al., 2021a), in which no $NO_x$ was present. This experiment intended to elucidate the ozonolysis chemistry. The chamber roof was closed at all times. In the first half of the experiment, about 4 ppbv of limonene was injected in addition to 45 ppbv $O_3$. After three hours about 4.5 ppbv of limonene was re-injected together with $O_3$, so that $O_3$ mixing ratios reached 70 ppbv. 100 ppmv of CO was added 30 minutes before the

second injection to scavenge OH radicals that are produced from the limonene ozonolysis reaction.






**Table 2.** Experimental conditions during the limonene oxidation experiments. $k_{bi}$ is the sum of the total loss rate of $RO_2$ due to bi-molecular reactions with NO, $HO_2$, and other $RO_2$ calculated from measured concentrations. Values are averages except for limonene concentrations, for which maximum concentrations after the injection are given. In experiments with multiple limonene injections, the range of the injections is noted.

| Experiment | [NO] (ppbv) | [$HO_2$] ($10^8$ cm$^{-3}$) | $k_{bi}$ (s$^{-1}$) | [OH] ($10^6$ cm$^{-3}$) | [$O_3$] (ppbv) | Limonene (ppbv) | Date |
|---|---|---|---|---|---|---|---|
| Ozonolysis | 0 | 1.5 – 6.0 | 0.005 – 0.15 | < 1 | 40 – 65 | 4 | 05 June 2020 |
| Low NO | 0.05 – 0.10 | 5 – 6 | 0.02 – 0.03 | 2 – 4 | 40 – 50 | 2 – 4 | 01 September 2012 |
| | 0.1 | 10 | 0.04 | 3 | 105 | 4 | 13 June 2015 |
| | 0.10 – 0.15 | 4 – 6 | 0.03 – 0.04 | 5 – 10 | 60 | 2 – 2.5 | 04 July 2019 |
| Medium NO | 0.2 | 3 | 0.05 | 2 | 5 | 4 | 08 August 2012 |
| | 0.3 | 4 | 0.07 | 4 | 5 | 4 | 10 August 2012 |
| High NO | 0.7 | 11 | 0.15 | 8 | 45 | 10 | 03 August 2015 |

## 2.4 Model calculations

The acquired measurements of trace gases and radicals are compared against a zero-dimensional box model applying the Master Chemical Mechanism (MCM) version 3.3.1 (Saunders et al., 2003; Jenkin et al., 2015). In addition to the chemistry from the MCM, chamber-specific processes including dilution and small productions of HONO, acetone and formaldehyde in the presence of sunlight are included in the model. Dilution in the chamber is implemented as a first-order loss process. The rate constant is calculated based on the monitored replenishment flow rate. The parameterization for the production of chamber sources for nitrous acid, formaldehyde and acetone follows the description in Rohrer et al. (2005); and Kaminski et al. (2017), in which production rates are parameterized as functions of temperature, relative humidity, and radiation. The source strength for the production rates are scaled for each experiment from the observed increase in concentrations for the part of the experiment when the chamber roof was opened, but limonene was not present (zero-air phase).

In addition, a background OH reactivity in the range of 1 s$^{-1}$ is present in the illuminated, clean chamber due to the presence of unmeasured OH reactants. In order to account for this background reactivity, an artificial OH reactant that behaves like CO is implemented in the model. Its concentration is adjusted in the model to match the observed OH reactivity during the zero-air phase and is assumed to be constant during the rest of the experiment.

The OH yield from limonene ozonolysis in the model calculations in this work is updated based on the IUPAC recommendations (Cox et al., 2020) decreasing the value from 87% to 66%. This is supported by multiple experimental studies (Aschmann et al., 2002; Herrmann et al., 2010; Forester and Wells, 2011) as well as theoretical studies (Wang and Wang,





2021). The decomposition product of the Criegee intermediate (MCM name: LIMOOB) is also updated, in which the terminal carbonyl group remains during the vinyl hydroperoxide mechanism (VHP). In addition, instead of forming a primary $RO_2$ (MCM name: C923O2) by eliminating a carbonyl group, decomposition of LIMOOB leads to the production of a secondary β-oxo substituted $RO_2$ (L5O2, Fig. 1). This is supported by the theoretical investigation of the ozonolysis of cyclohexene that the departure of the carbonyl group is not competitive without any inducing functional group (e.g., β-hydroperoxyl) (Rissanen

et al., 2014).

Three simulation model runs are performed for each experiment. In all model runs, physical parameters including temperature, pressure, photolysis frequencies, and the dilution rate of trace gases due to the replenishment flow are constrained to observations. If available, HONO concentrations are prescribed as measured to constrain the production from the chamber. In addition, ozone concentrations in the model are constrained to ensure that shortcomings of the model to predict ozone do not

complicate the interpretation by an inappropriate fraction of limonene reacting with ozone. Time series of NO and $NO_2$ concentrations are constrained to observations in the model runs except for the third set of simulations to avoid potential impacts of shortcomings of the model to describe these species.

In the first set of simulations (denoted as "reference run"), model runs are performed without any further constraints or modifications. In the second set of simulations (denoted as "constrained run"), $HO_2$ concentrations are prescribed as measured

to constrain the OH-production rate from the $HO_2+NO$ reaction. In this model run, OH reactivity is also adjusted to the measurements to yield an OH loss rate as observed (Section 3.3.1 to 3.3.4). In the third set of simulations (Section 3.5), NO and $NO_2$ concentrations are not constrained to investigate the fate of nitrogen oxides, radical concentrations and OH reactivity are also free parameters in the model.

OH reactivity from oxidation products tends to be overestimated by the model. Potential reasons include wall loss reactions of

low volatility compounds such as organic nitrates and peroxides, uncertainties from the reactivity of products and intermediates that are mostly derived using structure-activity relationship (SAR) in the model (Jenkin et al., 1997; Saunders et al., 2003), and chemical loss due to reactions that are not included in the MCM (e.g., isomerisation reactions). To reduce the OH reactivity in the model, it is assumed that organic nitrates and peroxides derived from limonene oxidation undergo a first order loss process.

The additional loss rate for organic nitrates is estimated from the decaying signal at the corresponding mass/charge ratio of the least oxidized C10 nitrate species ($C_{10}H_{17}NO_6$) observed by the chemical ionization mass spectrometry (CIMS) instrument after the chamber roof was closed in the experiment with high NO concentrations. The lifetime of the least oxidized species C10 nitrate is used to estimate the loss rate, as the only C10 nitrate species from limonene oxidation that the MCM includes is $C_{10}H_{17}NO_4$. The loss rate constant is equivalent to a lifetime of about 2 hours (Supplementary materials Fig. S1, Zhao et al.

(2018)), which is comparable to the chemical loss rate in the reaction with OH for conditions of these experiments (chemical lifetime of 0.5 to 4 hours). The specific reason for the additional loss of nitrates could not be identified, but might be due to



wall loss or due to loss by hydrolysis reactions in the humidified air similar to findings in an isoprene-rich forest (Romer et al., 2016).

Organic peroxides, however, could not be detected by the CIMS instrument and hence the additional loss rate is adjusted such
that the simulated OH reactivity agrees with measurements in the experiments with low and medium NO, when there should have been significant production of organic peroxides. This requires a lifetime of 10 minutes, much shorter than the lifetime of the order of hours under typical atmospheric conditions or those in the current experiments.

## 3. Results

### 3.1 Product yields from the oxidation of limonene

#### 3.1.1 Formaldehyde and acetone yields from the oxidation of limonene by OH and $O_3$

Formaldehyde (HCHO) is one of the oxidation products of the reaction of limonene with the OH radical as well as with ozone. It is mainly produced from the oxidation of the terminal double bond in limonene (Fig. 1). The product yield of HCHO from the oxidation of limonene by OH and $O_3$ can be determined from the chamber experiments by comparing the measured
concentrations of HCHO to the amount of oxidized limonene calculated from the measured limonene, OH and $O_3$ concentrations. As HCHO has also a small chamber source (Section 2.3) and it is lost by its reaction with OH and by photolysis, its concentration needs to be corrected for these losses following the method presented in Kaminski et al. (2017) and Rolletter et al. (2019).

For the evaluation of the HCHO yields, only measurements from the first two hours of the experiment after the first limonene
injection are used, because further oxidation of accumulating organic products could contribute to the production of HCHO at later times of the experiment. This is demonstrated by a rapid increase of HCHO when 80% of limonene had reacted away after a roughly linear relationship between HCHO and limonene concentrations (Fig. 1 and Supplementary Material Fig. S2). To derive the HCHO yield, a linear regression is performed for data until 40% of limonene is consumed and the slope of the linear regression gives the HCHO yield This results in values of (12±3)%, (13±3)%, and (32±5)% for the OH oxidation
experiments with low, medium and high NO mixing ratios, respectively (Fig. 2).

Formaldehyde is only produced from one of the three major $RO_2$ species formed in the initial reaction of limonene with OH. The structure-activity relationship (SAR) (Peeters et al. 2007) gives a yield of the formation of this $RO_2$ radical (Fig. 1) that results from the OH addition to the terminal C=C double bond (MCM name: LIMCO2) of 37% with an error of 15%. When the LIMCO2 radical reacts with NO or $RO_2$ radicals, an alkoxy radical is formed that subsequently forms HCHO from its
decomposition together with a carbonyl compound (MCM name: LIMKET, Fig. 1). Therefore, the expected HCHO yield





would be 37% at maximum from which the fraction of $RO_2$ reacting with other reaction partners such as $HO_2$ and the formation of organic nitrates in the reaction with NO (value determined in this work: (34±5) %, Section 3.1.2) needed to be subtracted. The uncertainty in the yield of organic nitrates could be higher because the values are determined for all $RO_2$ species from limonene oxidation by OH, but the nitrate yield for individual $RO_2$ could vary. The fraction of $RO_2$ reacting with NO is more

than 85% in the experiments with high and medium NO, and 65% in the experiment with low NO. The relative error of the fraction of $RO_2$ reacting with NO could be up to 35% due to the uncertainties of the reaction rate constants. Therefore, the HCHO yields expected from SAR in the experiments with high and medium NO is (22±9) % and in the experiment with low NO is (16±6) %.

All observed HCHO yields agree with SAR-derived HCHO yields which carry high relative uncertainties (40%). This could

explain the discrepancies between the observed yield for the experiment with high NO mixing ratios and the SAR-derived yield. The largest discrepancy could be observed for medium NO levels. A higher yield of LIMCO2 or lower yield of organic nitrates for the reaction of LIMCO2 with NO could explain the discrepancies observed for the experiments with low NO. Another possible reason is that there are other reaction channels of the LIMCO2 peroxy radical that are competitive with the bimolecular reactions with NO and $HO_2$ ($k_{bi}$ up to 0.07 s$^{-1}$) for conditions of the experiments with low and medium NO, but

would not be relevant in the experiment with high NO mixing ratios ($k_{bi}$ up to 2 s$^{-1}$). These reactions channels would not produce formaldehyde such as isomerisation. Possible isomerisation reactions of LIMCO2 are further discussed in Section 4.3.1.

The formation of HCHO at a later time in the experiments includes the production from further oxidation of first generation products. Therefore, HCHO yields are increasing over the course of the experiment to values of 40 to 90% with lowest yields

in the experiments with low NO (Fig. S2). Part of the differences between the experiments could be due to the differences in the contributions of various product species, so that numbers are not necessarily comparable. The HCHO yield remains below 100% in all experiments. This is consistent with HCHO being only produced from the oxidation of the terminal C=C bond of the limonene structure on a short time scale of serval hours in typical atmospheric conditions, so that total HCHO yield is limited to one in the limonene oxidation scheme.

Previous studies reported formaldehyde yields for the OH-oxidation of limonene between 36 and 43% (Larsen et al., 2001; Lee et al., 2006) (Table 3). The experiments by Larsen et al. (2001) and Lee et al. (2006) were performed at high NO concentrations, but the yield of HCHO yield is higher than the HCHO yield determined in the experiment with high NO in this work. However, experiments by Larsen et al. (2001) and Lee et al. (2006) were performed with much higher concentrations of limonene and NO (Table 3) and it is not clear, if only HCHO from the first oxidation step is considered (Fig. 1), so that

numbers may not be entirely comparable. In the study by Librando and Tringali (2005), formaldehyde yields were determined to be 27 and 92% in experiments with high limonene mixing ratios of 13 and 2 ppmv, respectively, but in the absence of NO. Yields in their experiments may not comparable to yields in this work, because the fate of organic peroxy radicals was likely dominated by atmospherically not relevant $RO_2+RO_2$ recombination reactions.



The HCHO yield derived from the pure ozonolysis experiment in this work is $(10\pm1)\%$ in the presence of the OH scavenger.
The same value is obtained, if the formaldehyde yield is determined from the part of the ozonolysis experiment without OH
scavenger, when approximately 40% of limonene reacted with OH that is produced from the ozonolysis reaction. Similar to
the OH reaction, formaldehyde production is expected from the subsequent chemistry after the ozone addition to the terminal
C=C double bond. Because ozone preferably (87%) adds to the endocyclic C=C double bond (Wang and Wang, 2021), a low
yield can be expected. As there was no NO present during the experiment, production of formaldehyde from the reaction
between limonene and OH could only be possible from the reaction of LIMCO2 with peroxy radicals or unimolecular reactions
(e.g. isomerisation). As the HCHO yield is the same independent on the addition of the OH scavenger, the yield from the
limonene-OH reaction at zero NO must be similar to the HCHO yield from the limonene ozonolysis.

The formaldehyde yield from the limonene ozonolysis derived in this work agrees well with the yield of 10% determined in
the work by Grosjean et al. (1993) performed with an OH scavenger. Yields were also determined in the work by Gong et al.
(2018) from several experiments at various chemical conditions. Values of 5 to 11% derived in their experiments with a
limonene:$O_3$ concentration ratio of 1:2 are consistent with results in this work. However, the HCHO yield was 11 to 27% in
the experiments with a limonene:$O_3$ concentration ratio of about 1:100. This higher yield might be explained by additional
formaldehyde production from the ozonolysis of secondary products (Fig. 1). It is worth noting that no significant differences
could be observed in experiments with and without OH scavenger in the experiments by Gong et al. (2018) similar to findings
in this work.

In the experiments in this work, there was no significant acetone production from the oxidation of limonene, as the production
rate of acetone from the chamber source ($\sim 80$ pptv h$^{-1}$) could already explain the observed increase of the acetone concentration
during the experiments. This suggests that unlike other monoterpenes, acetone is not a significant product from limonene
oxidation. This is consistent with findings by Lee et al. (2006) and Larsen et al. (2001). Also, the MCM does not predict
acetone as a product in the limonene oxidation scheme.





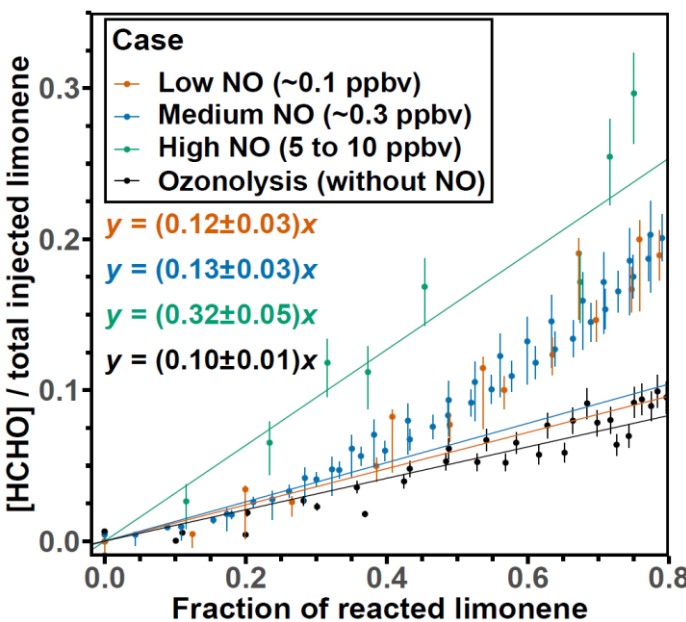

**Figure 2.** HCHO concentrations divided by the injected limonene concentration plotted versus the fraction of reacted limonene for the first injection in the experiments with different NO levels. The regression includes data points up to 40% of the fraction of reacted limonene.





**Table 3.** HCHO yields from the OH- and $O_3$-oxidation of limonene determined in the experiments in this work and reported in literature. Maximum values for limonene, $O_3$, and NO concentrations in the experiments are listed.

| Limonene + OH | HCHO yield (%) | Limonene (ppbv) | NO (ppbv) | |
|---|---|---|---|---|
| Larsen et al., 2001 | $36 \pm 5$ | 1000 | 1000 | |
| Librando and Tringali, 2005 | $27 - 92$ | $2100 - 13200$ | 0 | |
| Lee et al., 2006 | $43 \pm 5$ | 120 | 132[a] | |
| This work | $12 \pm 3$ | 3.5 | $0.05 - 0.15$ | |
| | $13 \pm 3$ | 3.5 | 0.3 | |
| | $32 \pm 5$ | 10 | $3 - 10$ | |
| Limonene + $O_3$ | HCHO yield (%) | Limonene (ppbv) | $O_3$ (ppbv) | OH scavenger |
| Grosjean et al., 1993 | 10 | 1200 | $70 - 100$ | Yes[b] |
| Gong et al., 2018 | $7 - 11$ | 280 | 500 | No |
| | $5 - 8$ | 280 | 500 | Yes[c] |
| | $13 - 27$ | 183 | 19000 | No |
| | $11 - 23$ | 183 | 19000 | Yes[c] |
| This work | $10 \pm 1$ | 4 | 50 | No |
| | $11 \pm 1$ | 4 | 80 | Yes[d] |

[a]Given as $NO_x$ concentration [b]200 ppmv cyclohexane [c]400 ppmv of 2-butanol or cyclohexane [d]60 ppmv CO

### 3.1.2 Organic nitrate yield from the oxidation of limonene by OH

The organic nitrate yield of the $RO_2$+NO reaction for peroxy radicals produced from the limonene+OH reaction is determined for experiments in this work following the method described for the analysis of previous experiments in the SAPHIR chamber (Hantschke et al., 2021; Tan et al., 2021). The amount of organic nitrates ($RONO_2$) in the chamber is estimated by the difference between the accumulated production of nitrogen oxides from the chamber source of HONO ($Q_{HONO}$), which is the only relevant source for nitrogen oxides in the experiments, and the total $NO_y$ (=$NO_x$ + HONO + $HNO_3$) concentrations. This method assumes that the differences can be solely attributed to the production of peroxy acyl nitrate (PAN)-like species, which can act as a reservoir for nitrogen oxides and can be significant in the oxidation of specific organic compounds.

The source strength of HONO can be calculated from the photo-stationary state between HONO, NO and OH using measured concentrations and measured photolysis frequencies:

$$\frac{d[\text{HONO}]}{dt} = Q_{\text{HONO}} - j_{\text{HONO}}[\text{HONO}] + k_{\text{OH+NO}}[\text{OH}][\text{NO}] \approx 0 \tag{1}$$



$$Q_{\text{HONO}} = j_{\text{HONO}}[\text{HONO}] - k_{\text{OH+NO}}[\text{OH}][\text{NO}] \tag{2}$$

The total amount of organic nitrates can be calculated from the difference between the accumulated HONO production and the sum of other nitrogen oxide species considering also their loss due to dilution in the chamber with rate $k_{\text{dil}}$ and the loss of $NO_2$ in the reaction with OH (Hantschke et al., 2021):

$$[\text{RONO}_2] = \int (Q_{\text{HONO}} - k_{\text{OH+NO}_2}[\text{NO}_2][\text{OH}] - k_{\text{dil}}([\text{NO}] + [\text{NO}_2] + [\text{HONO}]))dt - ([\text{NO}] + [\text{NO}_2] + [\text{HONO}]) \tag{3}$$

Assuming that at least right after the injection of limonene, the production of organic nitrates is dominated by the reaction of NO with $RO_2$ derived from limonene-OH, the organic nitrate yield $\Phi_{\text{limonene+OH}}$ can be calculated:

$$[\text{RONO}_2] = \Phi_{\text{limonene+OH}} \int ([\text{RO}_2]_{\text{limonene}} \times [\text{NO}] \times k_{\text{RO}_2+\text{NO}}) \, dt \tag{4}$$

The fraction of $RO_2$ from limonene-OH oxidation to the measured total $RO_2$ concentration is estimated based on the ratio of
the OH reactivity from limonene to the OH reactivity from all OH reactants that are expected to produce $RO_2$ in the reaction with OH (Tan et al., 2021). The latter can be calculated by subtracting the background OH reactivity ($k_{\text{OH}_{\text{bg}}}$) that includes the reactivities attributed to inorganic species and formaldehyde from the total measured OH reactivity ($k_{\text{OH}}$):

$$[\text{RO}_{2_{\text{limonene}}}] = \frac{k_{\text{OH}_{\text{limonene}}}}{k_{\text{OH}} - k_{\text{OH}_{\text{bg}}}} [\text{RO}_2] \tag{5}$$

Using Equation 3 and 4, the nitrate yield $\Phi_{\text{limonene+OH}}$ is obtained as the slope of the linear regression between the calculated
organic nitrate concentrations and the integrated turnover rate of the reaction between $RO_{2_{\text{limonene}}}$ and NO. Only experiments with medium NO mixing ratios are used for this analysis here, as HONO measurements were performed and more than 90% of the limonene was oxidized by OH in these experiments. This results in a $\Phi_{\text{limonene+OH}}$ of $(34\pm5)\%$ (Fig. 3). The uncertainty of this calculation is mainly due to the accuracy of concentration measurements of $NO_y$ species and radicals. The organic nitrate yield of the $RO_2$ from limonene ozonolysis is not determined in this study as a considerable amount of PANs could be
formed according to the MCM model. Therefore the organic nitrate yield analysis is not conducted for the experiments at low NO mixing ratio.

The organic nitrate yield in this study is higher than values estimated using different SAR approaches of 19% (Jenkin et al., 2019) to 28% (Arey et al., 2001; Leungsakul et al., 2005). However, the value agrees well with nitrate yield derived in experiments, in which the yield was derived from the analysis of the aerosol chemical composition produced from the oxidation
of limonene ($(36\pm6)\%$; Rollins et al. (2010)).



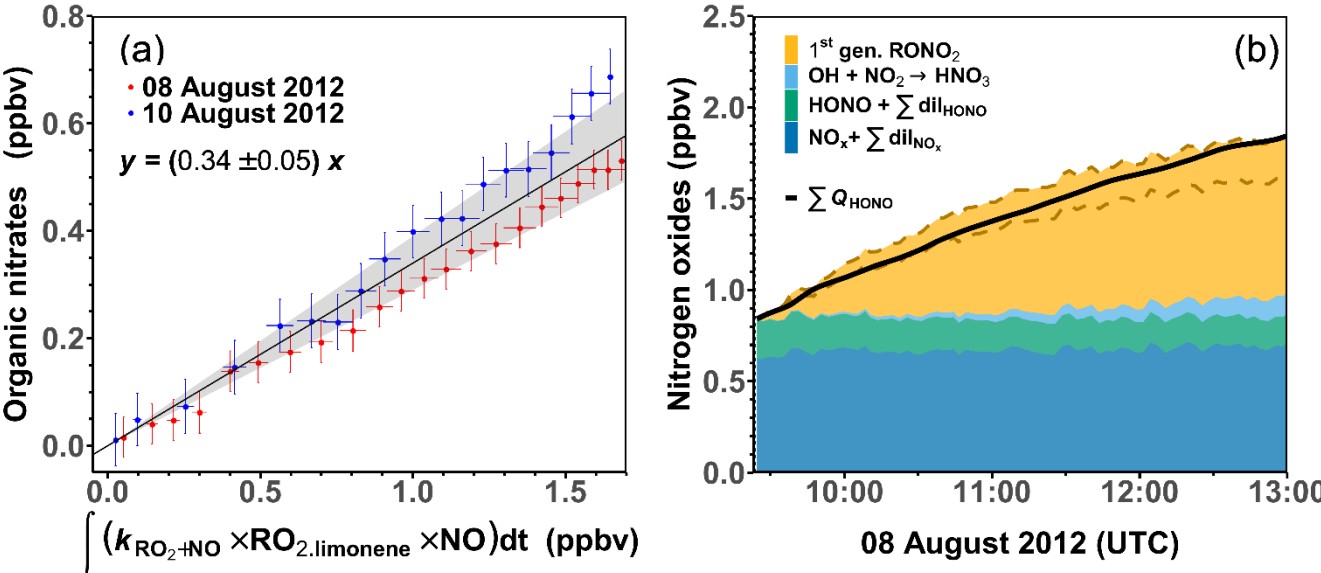

**Figure 3.** (a) Regression between calculated organic nitrate mixing ratio and integrated turnover rate of the reaction of limonene-derived $RO_2$ and NO. The regression only includes data when less than 60% of the limonene from the first injection reacted away. (b) Concentrations of nitrogen oxide species after the first limonene injection in the experiment with medium NO on 08 August 2012 compared to the total production of HONO from the chamber source. The organic nitrate contribution is calculated using a nitrate yield of (34±5)% for the reaction of limonene $RO_2$ with NO, the uncertainty range is enclosed by the two dashed lines.

## 3.2 Comparison of modelled and measured concentration time series

### 3.2.1 Model-measurement comparison for limonene photo-oxidation experiments at low NO mixing ratio

Three experiments were performed at low NO concentrations on 01 September 2012 (Fig. 4), 13 August 2015 and 04 July 2019 (Supplementary Material, Fig. S3 and S4). Measured OH concentrations during the limonene oxidation were about (2-7)×$10^6$ cm$^{-3}$ in the experiments in 2012 and about (5-11)×$10^6$ cm$^{-3}$ in the experiments in 2019. OH concentrations were highest, when most of the limonene was consumed. The overall higher OH concentrations in the experiments in 2019 were due to the higher NO mixing ratios (~0.15 ppbv) compared to the experiment in 2012 (~0.05 ppbv), which led to a faster OH production from the radical regeneration reaction of $HO_2$ and NO. In these experiments, around 30 to 40% of limonene reacted with $O_3$ and 60 to 70% of limonene reacted with OH. Measured $HO_2$ concentrations were around (5-8)×$10^8$ cm$^{-3}$. Measured $RO_2$ concentrations were around (2-5)×$10^8$ cm$^{-3}$ in the experiment in 2012 and (5-10) ×$10^8$ cm$^{-3}$ in the experiments in 2019, respectively. The average chemical lifetime of $RO_2$ to bimolecular reactions with NO, $HO_2$ and $RO_2$ can be calculated using





measured NO, $HO_2$ and $RO_2$ concentrations, respectively, resulting in a chemical lifetime of 25 to 50s. 60 to 70% of the $RO_2$ radicals reacted with NO and 30 to 40% reacted with $HO_2$. Loss due to $RO_2+RO_2$ recombination reactions are predicted not to play a major role, if reaction rate constants are taken from the MCM that are much lower compared to the rate constants of the reactions with $HO_2$.

In the reference model run, which essentially uses the MCM chemistry without constraining radical production or destruction processes (Section 2.4), the OH reactivity is overestimated in the model by 3 to 5 $s^{-1}$ after nearly all limonene has reacted away. Right after the limonene injections, $HO_2$ concentrations are underestimated by 20 to 50% and the $RO_2$ concentrations are overestimated by 500 to 700%. OH is underestimated by at least 50%, which could partly result from the overestimation of the OH reactivity and underestimated $HO_2$ concentration (and thereby the reaction rate of $HO_2+NO$). The underestimated OH
concentration also leads to a slightly slower decay of the modelled limonene, so that modelled limonene concentrations are about 0.1 – 0.2 ppbv higher than measured values at the end of the experiment.

    In the model run with adjusted OH reactivity and constrained $HO_2$ concentration (constrained model run), the discrepancy between modelled and measured OH concentrations is reduced. They are still underestimated by 33% on average throughout the whole experiment and by about 50% one hour after the limonene injection, which could indicate a missing OH production
process in the model. However, the temporal behaviour of the modelled limonene concentration agrees with the behaviour of the measurements, suggesting that measured OH concentrations are too high.

    Maximum modelled $RO_2$ concentrations are reduced by about 30% compared to the reference run as a result of the increased $HO_2$ concentration in the constrained run leading to a higher $RO_2$ loss by $RO_2+HO_2$ recombination reactions. However, this improvement is insufficient to explain measured $RO_2$ concentrations, which are still at least a factor 4 lower than modelled
values.



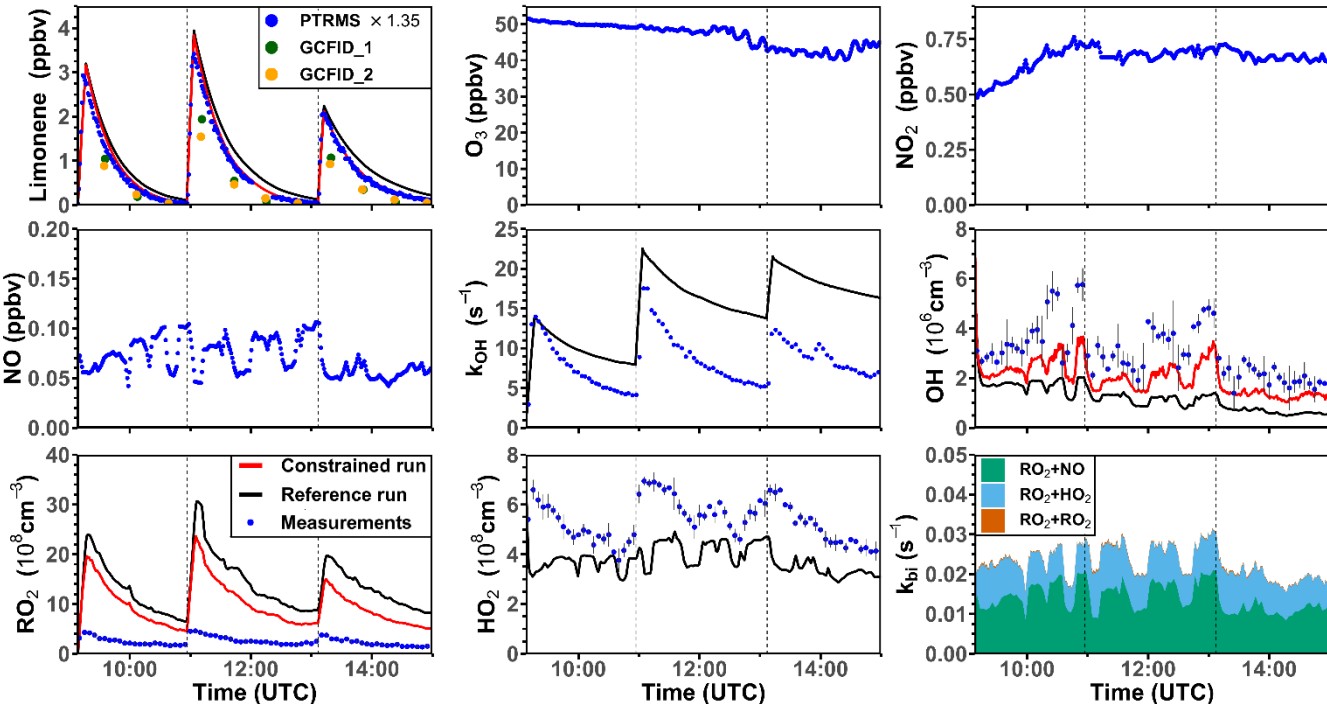

**Figure 4** Time series of radicals, inorganic and organic species in the limonene oxidation experiment with low NO mixing ratios on 01 September 2012. Limonene concentrations measured by the PTRMS are scaled by a factor of 1.35 to match the increase of OH reactivity during the injections of limonene. The vertical dashed lines represent times, when limonene was

injected into the chamber. $RO_2$ bi-molecular reaction loss rates ($k_{bi}$) are calculated based on the measured NO, $HO_2$, and $RO_2$ concentrations using the reaction rate constants as included in the MCM. In the constrained model run $HO_2$ concentrations are constrained to measurements and the OH reactivity is adjusted by additional loss reactions to match measurements.

**3.2.2 Model-measurement comparison for limonene photo-oxidation experiments at medium NO mixing ratios**

Two oxidation experiments were performed at medium NO levels on 08 and 10 August 2012 (Fig. 5 and Supplementary Material Fig. S5). Throughout both experiments, measured NO mixing ratios ranged from 0.25 to 0.4 ppbv. More than 90% of the $RO_2$ derived from the limonene-OH reaction reacted with NO. $O_3$ mixing ratios increased gradually from about 1 ppbv to 10 ppbv as a result of the photolysis of $NO_2$, which was produced from the reaction between organic peroxy radicals and $HO_2$ with NO. At such low $O_3$ concentrations, the reaction with OH was the dominant loss pathway for limonene (90%). The

measured OH concentration ranged from $(2-5) \times 10^6$ cm$^{-3}$ and the measured $HO_2$ and $RO_2$ concentrations were about $(2-5) \times 10^8$ cm$^{-3}$.





In the reference model run, the simulated OH reactivity increasingly deviates from measurements over the course of the experiment. The differences are about 2 to 4 $s^{-1}$ after two to three hours of oxidation when all limonene has been consumed. Contributors to the OH reactivity in the model in addition to limonene are mostly oxidation products such as aldehydes and ketones (~ 70%), organic peroxides (~ 15%) and organic nitrates (~ 15%) (Supplementary Material, Fig. S6). Simulated $HO_2$ concentrations are underestimated in the reference model run by around 10 to 30%. Hence, the simulated OH concentration is underestimated by 30 to 80% due to the slow regeneration rate from the reaction between $HO_2$ and NO as well as the faster removal rate of OH. As a result, limonene concentrations are overestimated by the model up to 0.3 ppbv throughout the experiment.

In the constrained model run, the simulated OH concentration is on average 20% lower than measured values. This difference is within the uncertainties of OH measurements. The higher modelled OH concentrations compared to the reference model run leads to a faster consumption of limonene compared to the reference model. This temporal behaviour better agrees with the temporal behaviour observed by PTRMS measurement, but the limonene consumption is slightly faster in the model than measurements suggests. Modelled $RO_2$ concentrations are 50 to 100% higher than measurements. Values are 10 to 20% higher compared to the reference model run right after the injection of limonene presumably due to the enhanced $RO_2$ production in the constrained model run, in which OH concentrations are higher.



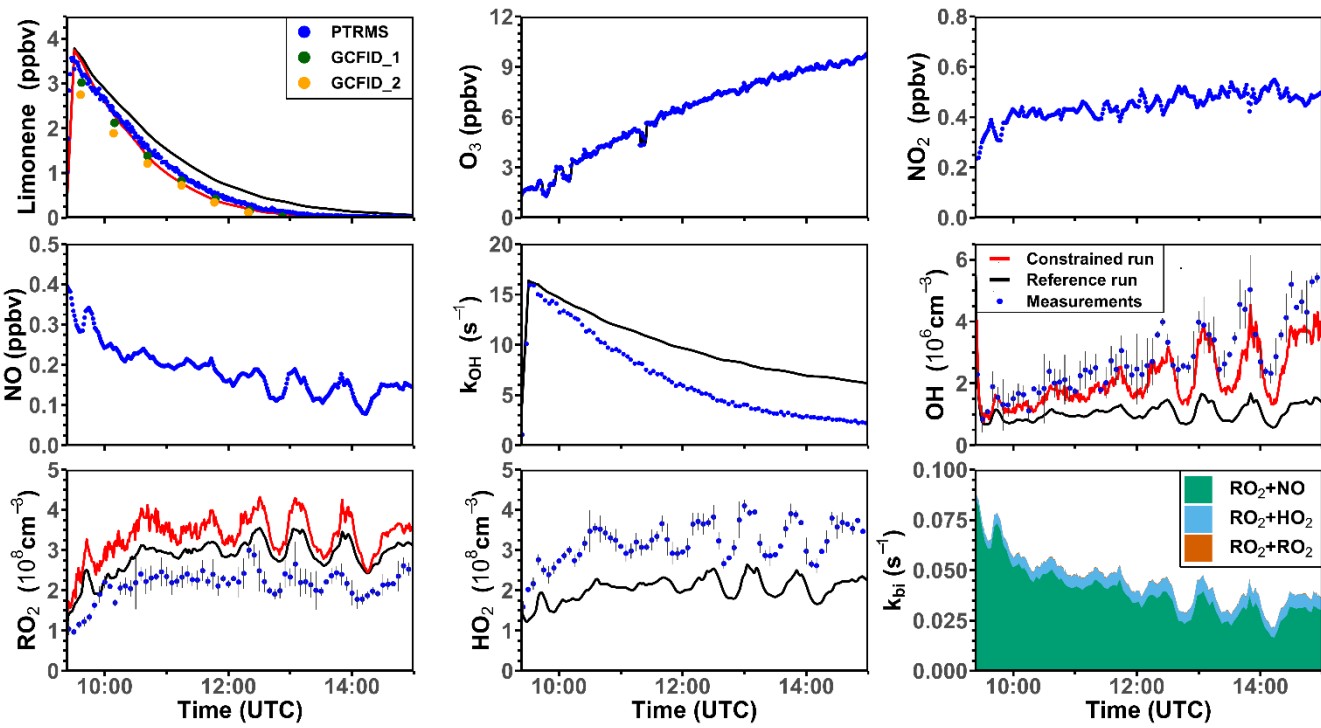

**Figure 5.** Time series of radicals, inorganic and organic species in the limonene oxidation experiment at medium NO level on 08 August 2012. $RO_2$ bi-molecular reaction loss rates ($k_{bi}$) are calculated based on the measured NO, $HO_2$, and $RO_2$ concentrations using the reaction rate constants from the MCM. In the constrained model run $HO_2$ concentrations are constrained to measurements and the OH reactivity is adjusted by additional loss reactions to match measurements.

### 3.2.3 Model-measurement comparison for limonene photo-oxidation experiment at high NO mixing ratios

One experiment was performed with high NO mixing ratios on 03 August 2015. In this experiment, limonene was injected twice. Unfortunately, measurements of radicals failed during the first part of the experiment, so that only the part of the experiment after the second limonene injection can be analysed (Fig. 6). The ozone mixing ratio at the beginning of this part of the experiment was about 35 ppbv as a result of ozone production during the first part of the experiment. The OH concentration measured by DOAS was around $(5-10) \times 10^6$ cm$^{-3}$, so that around 70% and 30% of limonene reacted with OH and $O_3$, respectively. The OH concentration measured by LIF was around $(5-15) \times 10^6$ cm$^{-3}$. The $HO_2$ concentration was about $1 \times 10^9$ cm$^{-3}$ throughout the experiment. The high NO and $HO_2$ concentrations led to a short $RO_2$ lifetime of 5 to 10 s, much shorter than in the other experiments with lower NO concentrations. The removal of $RO_2$ by bimolecular reactions was mostly due to the reaction with NO (85 %) and $HO_2$ (15 %).





In the reference model run, the model-measurement discrepancy of the OH reactivity is around 3 $s^{-1}$ before the second limonene injection, and it further increases to 10 $s^{-1}$ at the end of the experiment. The absolute discrepancy in OH reactivity is higher compared to other experiments because the total amount of limonene and therefore the production of oxygenated products was higher. Modelled OH and $HO_2$ concentrations are both lower by 50 to 70% than the measured values. The low OH
concentration in the model again leads to a slower decay of modelled limonene concentrations compared to observed values. $RO_2$ concentrations show a good agreement with the measurement, with a discrepancy of less than 20%.

In the constrained model run, the simulated OH concentration is in good agreement with the measurement by DOAS in the first hour and only slightly underestimates the measurements by about 20% at the end of the experiment. As a result, the agreement between modelled and measured time series of limonene concentrations improves compared to the reference model
run. Differences between $RO_2$ concentrations in the constrained and reference model are rather small except for the point in time when limonene is injected and $RO_2$ concentrations in the constrained model run increase more rapidly than observations. This results in 30% difference between modelled and measured values during the first hour of the experiment. This can be explained by the increased $RO_2$ production from the limonene oxidation by the OH radical which concentration is underestimated in the reference model run. Possible reasons for the overestimated $RO_2$ concentration could include an
underestimated loss rate of $RO_2$ at NO mixing ratio of about 1 ppbv (Section 3.4), or an overestimated production of $RO_2$ from the further oxidation of products from the previous limonene injection.



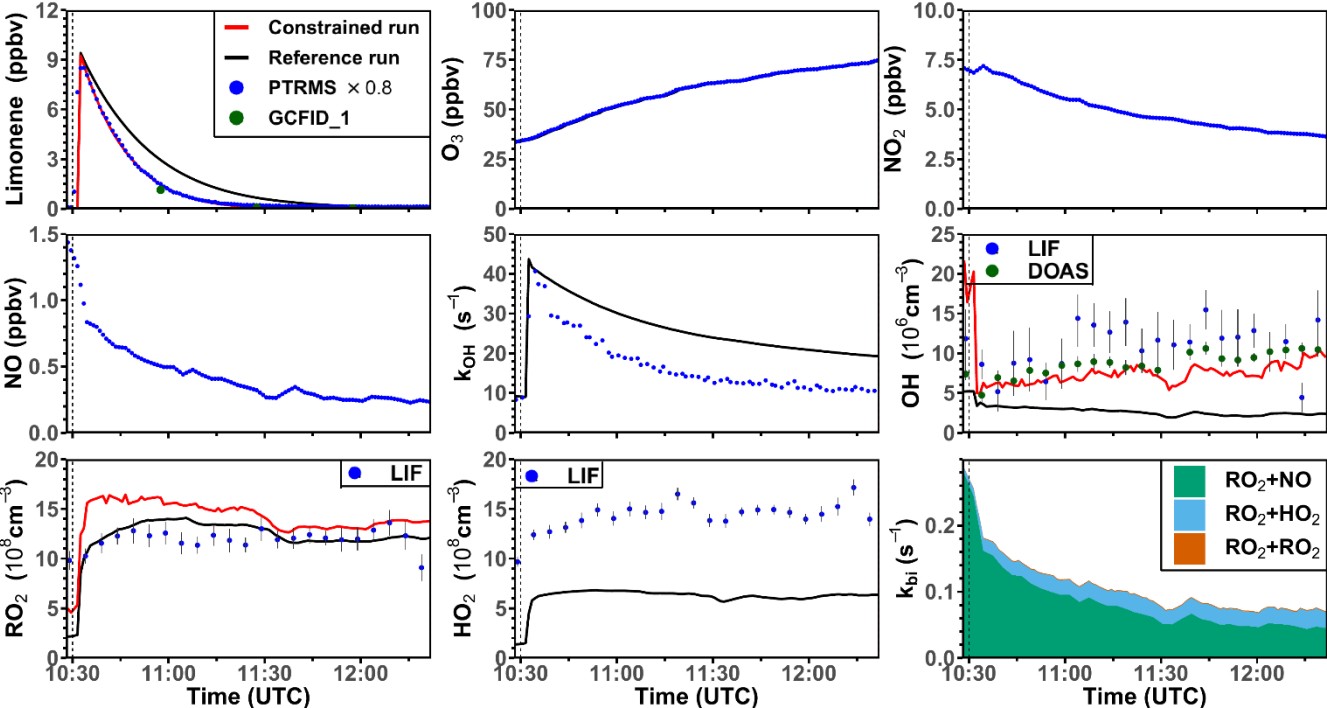

**Figure 6.** Time series of radicals, inorganic and organic species during the limonene oxidation experiment at high NO mixing ratios on 03 August 2015. Limonene concentrations measured by the PTRMS are scaled by a factor of 0.8 to match the increase of OH reactivity during the injection of limonene. $RO_2$ bi-molecular reaction loss rates ($k_{bi}$) are calculated based on the measured NO, $HO_2$, and $RO_2$ concentrations using the reaction rate constants from the MCM. In the constrained model run $HO_2$ concentrations are constrained to measurements and the OH reactivity is adjusted by additional loss reactions to match
measurements.

**3.2.4 Model-measurement comparison for limonene ozonolysis experiment in the dark with and without OH scavenger**

The ozonolysis of limonene in the dark was investigated in a separate experiment on 05 June 2020 (Fig. 7). A significant OH concentration was observed with values up to $1.2 \times 10^6$ $cm^{-3}$ after the first injection of limonene as a result of the OH production
from the ozonolysis reaction. OH concentrations were below the limit of detection of both DOAS and LIF after 100 ppmv CO had been injected as OH scavenger right before the second limonene injection. The measured $HO_2$ concentration was around $3 \times 10^8$ $cm^{-3}$ after the first limonene injection and increased to maximum concentrations of $1 \times 10^9$ $cm^{-3}$ after the second limonene injection due to the conversion of OH radicals formed from the ozonolysis in the reaction with excess CO. Because there was no limonene oxidation by OH, there was also less production of $RO_2$ radicals than the first limonene injection. However, the
observed maximum $RO_2$ concentration after each limonene injection was about $1 \times 10^9$ $cm^{-3}$.



Similar as observed in the other experiments, modelled OH reactivity is higher than measured values by 5 s$^{-1}$. The maximum discrepancies appear right before the injection of CO, after which OH reactivity could not be measured due to the high contribution of CO. In the reference model run, HO$_2$ is underestimated by the model by around 90% during the first part of the experiment and modelled HO$_2$ is also a factor of three lower than measurements during the pure ozonolysis part in the presence

of CO as OH scavenger. In contrast, measured RO$_2$ concentrations are drastically overestimated by the model by a factor of 7 right after each limonene injection.

In the constrained model run, simulated OH concentrations are 25% higher compared to the reference model as a result of the reduced OH reactivity in the model. The main difference between the results of the reference model run and the constrained model run is that RO$_2$ concentrations are reduced by half in the constrained model. This is caused by an increased RO$_2$ removal

rate by the high HO$_2$ concentration.

Using measured RO$_2$ and HO$_2$ concentrations, the maximum reaction rates of the reaction of RO$_2$ with HO$_2$ is about 0.005 s$^{-1}$ and 0.02 s$^{-1}$ for the first and second limonene injection, respectively. In contrast, the RO$_2$ loss rate due to the recombination of organic peroxy radicals is calculated to be only 0.001 s$^{-1}$, so that it is expected that RO$_2$ reacted mainly with HO$_2$ to form peroxides, if only bimolecular reactions are considered. However, the loss due to RO$_2$-RO$_2$ reactions could be of similar

importance as the loss due to RO$_2$-HO$_2$ reactions, if low HO$_2$ and high RO$_2$ concentrations as predicted by the reference model run are used for the calculation. The large discrepancies between measured and modelled RO$_2$ and HO$_2$ radical concentrations and possible explanations are further discussed in Sections 4.1 and 4.2.





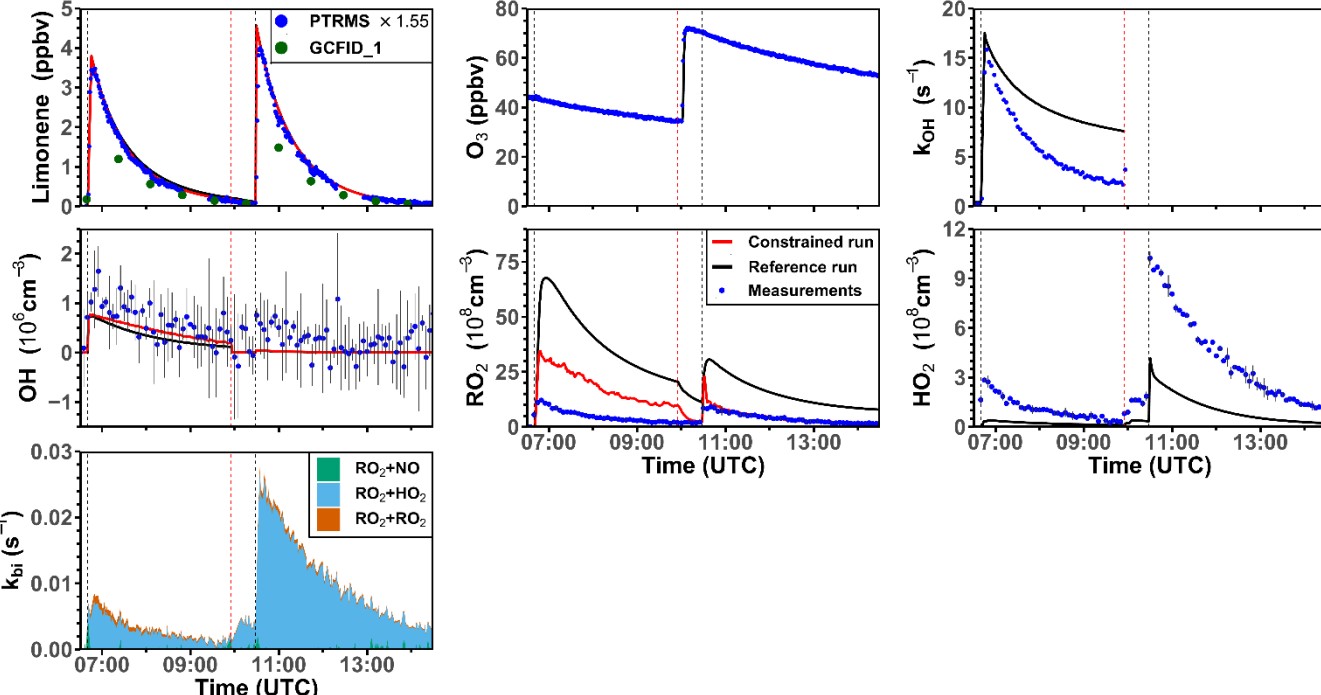

**Figure 7.** Time series of radicals, inorganic and organic species during the limonene ozonolysis experiment in the dark on 05 June 2020. Limonene concentrations measured by the PTRMS are scaled by a factor of 1.55 to match the increase of OH reactivity during the injection of limonene. The black vertical dashed lines represent when limonene was injected into the chamber; the red vertical dashed line represents the injection of 100 ppmv of CO. $RO_2$ bi-molecular reaction loss rates ($k_{bi}$) are calculated based on the measured NO, $HO_2$, and $RO_2$ concentrations using the reaction rate constants in the MCM. In the constrained model run $HO_2$ concentrations are constrained to measurements and the OH reactivity is adjusted by additional loss reactions to match measurements.

**3.3 Chemical OH radical budget using measured quantities**

To evaluate whether unaccounted chemical processes are required to explain the total OH destruction rate in the experiments, the rates of OH-producing reactions are calculated using measured trace gas and radical concentrations. Turnover rates of single reactions are summed up and compared to the total OH destruction rate. Due to the short chemical lifetime of OH radicals (< 1 s) steady-state conditions are expected that the production and destruction rates must be balanced.

The total OH destruction rate is obtained by computing the product of the measured OH reactivity and OH concentration. In experiments in year 2015, when OH was measured by both DOAS and LIF instruments, DOAS measurements are used for the





calculation of total OH destruction rate. The uncertainty of the OH destruction rate is 16% calculated by error propagation of
the measurement uncertainties. The main OH production processes included in the chemical budget analysis are listed in Table
4. OH production by photolysis of HONO and $O_3$, the reaction between $HO_2+NO$, and between $HO_2+O_3$, and the limonene
ozonolysis reaction are considered. The uncertainties of these OH-producing processes range from 15% (photolysis processes)
to 20% ($HO_2+NO$ reaction and limonene ozonolysis reaction), which are derived from error propagation of the uncertainties

of measurements (Table 1) and reaction rate constants (Atkinson et al., 2004; Cox et al., 2020).

Figure 8 and Figure S7 show the chemical budgets of OH radicals in the limonene oxidation experiments with different NO
mixing ratios. The destruction and production rates of OH for experiments with medium (08 August 2012) and high NO (03
August 2015) are balanced within the uncertainties of measurements during all experiments. This is consistent with the overall
good agreement between modelled and measured OH radical concentrations obtained for the constrained model run (Fig. 5

and 6). In the experiment with high NO, the total OH turnover rate is higher ($> 20$ ppbv h$^{-1}$) compared to the other experiments
due to the high limonene concentration (10 ppbv) and the high NO concentration, both of which accelerate the turnover of
radicals.

In experiments with medium and high NO concentrations, the main OH source after the limonene injection is the reaction
between $HO_2$ and NO ($>85\%$), followed by the photolysis of HONO in the experiment with medium NO, and ozone photolysis

and limonene ozonolysis in the experiment with high NO, when 60ppbv $O_3$ was present. In total, about 85 to 100% of the OH
radical production rate can be explained by the calculated processes that are listed in Table 4.

In the experiment with low NO on 01 September 2012 (Fig. 8b), the OH production rate is 6 ppbv h$^{-1}$ after the injection of
limonene, of which about 33% can be attributed to OH production from limonene ozonolysis. The NO concentration varies
over the course of the experiment making OH regeneration from the $HO_2+NO$ reaction an important OH source with a

contribution of 15 – 50% to the total OH production in addition to OH production from the photolysis of HONO and $O_3$. About
1.0 ppbv h$^{-1}$ (20 – 33%) of the OH destruction rate is not explained by these OH production processes, which is consistent with
the underestimation of OH concentrations in the constrained model run compared to measured values (Fig. 4). However, the
gap between the OH destruction rate and destruction rate is about 1 ppbv h$^{-1}$ throughout the whole experiment and does not
vary with the amount of limonene presence in the chamber. This suggests that either the missing OH source is not related to

the oxidation of limonene or is due to measurement artefacts for example in the OH measurements, which would lead to an
overestimation of the OH destruction rate. An artefact in the OH measurement would also be consistent with what is observed
in the constrained model run (Fig. 4), where the rapid decrease of the modelled limonene concentration suggests that the
measured OH is too high in this experiment.

In the ozonolysis experiment without OH scavenger (Fig. 8d), OH is only produced by ozonolysis reaction of limonene. The

total OH production rate is about 2 to 3 ppbv h$^{-1}$ at the beginning of the oxidation and gradually declines while limonene is





being consumed. The total OH destruction rate is well explained by the production from limonene ozonolysis suggesting that OH production from further ozonolysis reactions of product species is not significant for conditions of this experiment.

In conclusion, the OH production rate from the four major OH sources that are included in the calculations (Table 4) is balanced by the OH destruction rate within the 25% uncertainty of the calculation at the beginning of the experiment, when limonene-
oxidation is most important in the experiments with medium and high NO and in the ozonolysis experiment. In the experiments with low NO concentrations, imbalances of 20 to 33% are observed indicating that an additional OH production process with a rate of 1.0 ppbv h$^{-1}$ would be required to explain the observed destruction rate, but there are indications that this could be due to a measurement artefact in the OH measurements.

In all experiments, the OH production rates are lower than OH destruction rates at later times of the experiments, when
secondary chemistry becomes important. However, differences are similar to the uncertainty of the calculations. These discrepancies may indicate that additional OH could be produced from unaccounted reactions of oxidation products for example from the photolysis of organic peroxides in the photo-oxidation experiments (Badali et al., 2015).

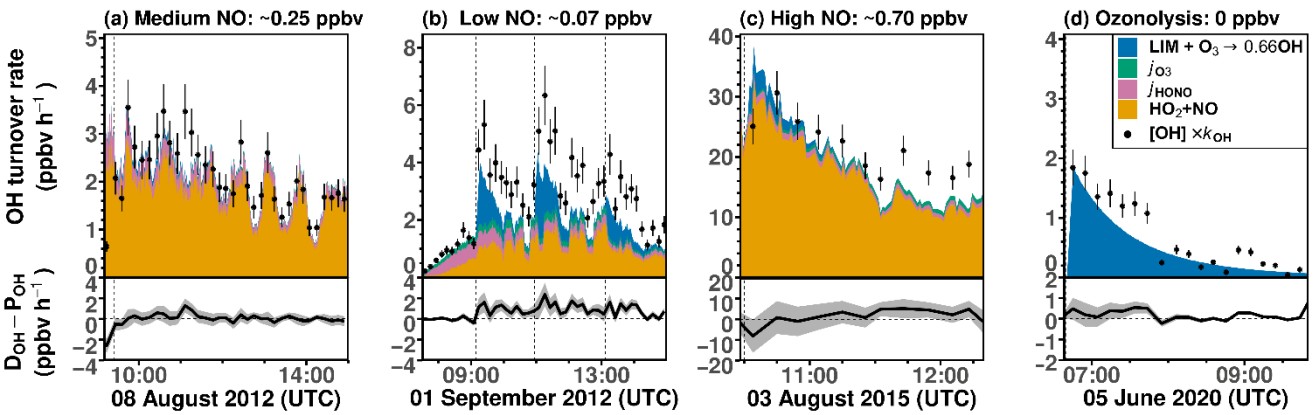


**Figure 8.** 10-minutes-average values of total OH destruction rates compared to the sum of OH production rates from the major OH sources that can be calculated from measurements in the (a) medium NO, (b) low NO, (c) high NO, and (d) ozonolysis experiments. Shaded areas in the difference plots give the uncertainties of the calculations. Production of OH from the reaction of $O_3$ and $HO_2$ is not included, because the contribution to the total OH production was negligible (< 0.01 ppbv h$^{-1}$) for
conditions of the experiments.





**Table 4.** Reactions that are included in the analysis of the OH production rate. Reaction rate constants are given for 298K and 1 atm. The reaction rate constants that are used in the analysis are calculated for the measured temperature.

| Reaction: | $k$ (298 K, 1 atm) | 1σ uncertainty of $k$ (%) | Reference: |
|---|---|---|---|
| $HO_2 + NO \rightarrow OH + NO_2$ | $8.8 \times 10^{-12}$ cm$^3$ s$^{-1}$ | 11 | Atkinson et al. (2004) |
| $HO_2 + O_3 \rightarrow OH + 2O_2$ | $2.0 \times 10^{-15}$ cm$^3$ s$^{-1}$ | 22 | Atkinson et al. (2004) |
| $HONO + h\nu \rightarrow OH + NO$ | $j_{HONO}$ | 10 | Measured |
| $O_3 + h\nu \rightarrow O(^1D) + O_2$ | $j_{O_3}$ | 10 | Measured |
| $O(^1D) + H_2O \rightarrow 2OH$ | $2.1 \times 10^{-10}$ cm$^3$ s$^{-1}$ | 11 | Atkinson et al. (2004) |
| $O(^1D) + M \rightarrow O(^3P) + M$ | $2.6 \times 10^{-11}$ cm$^3$ s$^{-1}$ | 11 | Atkinson et al. (2004) |
| $Limonene + O_3 \rightarrow 0.66RO_2 + 0.66OH$ | $2.1 \times 10^{-16}$ cm$^3$ s$^{-1}$ | 11 | Cox et al. (2020) |

## 3.4 Chemical budget of first-generation peroxy radicals using measured quantities

The discrepancies between measured and simulated organic peroxy radical concentrations are much higher (up to a factor of
2) in the experiments with low NO concentrations and in the ozonolysis experiment compared to the experiments with medium or high NO concentrations. The analysis of the composition of the $RO_2$ concentrations using model results from the constrained model run (Section 2.4) shows that the concentrations of $RO_2$ produced in the initial reaction of limonene with OH and $O_3$ already exceed the measured total $RO_2$ concentrations (Fig. 9 and Supplementary Material Fig. S8). Model-measurement percentage differences are at least a factor of 2 higher than the accuracy of the measured $RO_2$ concentration (~25%). Therefore,
the discrepancy suggests that additional loss pathways for $RO_2$ have to be included in the model.

To examine the magnitude of the additional $RO_2$ loss rate, a chemical budget analysis for $RO_2$ radicals is performed similarly to the analysis for OH radicals (Section 3.3). As the chemical loss rates of peroxy radicals are within the range of 0.01 to 0.20 s$^{-1}$, steady-state concentrations can be assumed. The production rate of the peroxy radicals produced right after the limonene injection is well-defined by the loss rate of limonene due to the reactions with OH- and $O_3$. Therefore, only measurements
during the first 30 minutes after the first limonene injection are used for the analysis, so that calculations are not impacted much by additional $RO_2$ production from the subsequent oxidation of organic products. The removal rate of peroxy radicals includes bimolecular reactions ($k_{bi}$):





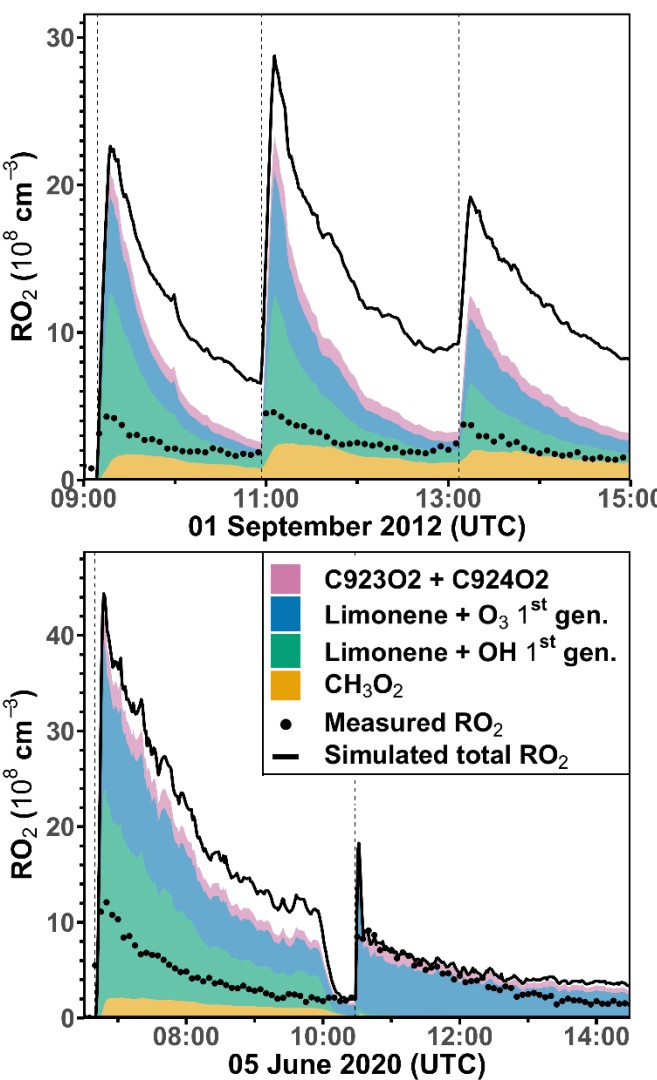

**Figure 9.** Total RO$_2$ radical concentrations and their speciation from model calculations (constrained model run) compared to the measured values (black dots) for the experiments with low NO and the ozonolysis experiment. Methylperoxy radicals (CH$_3$O$_2$) are mainly produced from the oxidation of HCHO. Radicals produced in the initial reactions of limonene with either OH or O$_3$ are summed. C923O2 + C924O2 are RO$_2$ radicals produced from the further oxidation of the first-generation oxidation products. Names are taken from the MCM model.





**Table 5.** Additional removal rates ($k_{add}$) that are required to balance the $RO_2$ production and destruction rates in the different experiments together with conditions of the experiments such as the percentage of limonene that reacted with OH (LIM+OH) or $O_3$ (LIM+$O_3$). Only data from 30 minutes after the first limonene injection is analysed.

| NO | Date | T (K) | LIM+OH (%) | LIM+$O_3$ (%) | $P_{RO2}$ ($10^7$ cm$^{-3}$ s$^{-1}$) | [$RO_2$] ($10^8$ cm$^{-3}$) | $k_{bi}$ ($10^{-2}$ s$^{-1}$) | $k_{add}$ ($10^{-2}$ s$^{-1}$) |
|---|---|---|---|---|---|---|---|---|
| Zero | 05 June 2020 | 286 | 49 | 51 | 2.3 ± 0.6 | 10.0 ± 1.4 | 0.9 ± 0.3 | 1.4 ± 0.7 |
| Low | 01 September 2012 | 313 | 59 | 41 | 2.8 ± 0.8 | 3.5 ± 0.5 | 2.5 ± 0.7 | 5.6 ± 2.7 |
| Low | 04 July 2019 | 302 | 72 | 28 | 3.1 ± 0.9 | 6.4 ± 1.0 | 3.1 ± 0.9 | 1.7 ± 1.7 |
| Medium | 08 August 2012 | 303 | 97 | 3 | 1.7 ± 0.5 | 1.4 ± 0.2 | 6.9 ± 2.1 | 5.2 ± 3.5 |
| Medium | 10 August 2012 | 302 | 95 | 5 | 1.8 ± 0.5 | 1.6 ± 0.2 | 8.2 ± 2.7 | 3.3 ± 4.2 |

### 3.5 Impacts of $RO_2$ model results on the modelled $NO_x$ concentrations

In the model runs described so far, NO and $NO_2$ concentrations are constrained to measured values. If these are not constrained, $NO_x$ is underestimated at the beginning of the experiments, but values are overestimated at the end (Fig. 10 and Supplementary Material Fig. S9). The discrepancy at the beginning can be mainly attributed to the overestimation of modelled $RO_2$ concentration, which leads to an overestimation of the formation of organic nitrates that act as sinks for $NO_x$ on the timescale of the experiments. To illustrate the impact of $RO_2$ concentrations on the modelled $NO_x$ concentrations, two model runs are

compared: one with modelled $RO_2$ concentrations (reference run) and the second with modelled $RO_2$ concentrations adjusted to match the measurements.

$RO_2$ concentrations are adjusted by applying an additional loss with a constant rate constant to all six first-generation $RO_2$ derived from -OH and -$O_3$ oxidation (Fig. 1). The additional loss rate for $RO_2$ is around 0.01 to 0.06 s$^{-1}$, similar to the loss rate derived in the analysis of the chemical budget for $RO_2$ ($k_{add}$, Section 3.4). In both model runs, the organic nitrate yield of the

$RO_2$+NO reaction for the first-generation $RO_2$ radicals from limonene+OH is adjusted to the yield (34%) that is found in the analysis of the experiments at medium NO mixing ratio (Section 3.1.2). Measured $j_{NO2}$, $O_3$ and $HO_2$ concentrations are used in both model runs to constrain the loss rate of $NO_2$ and NO. It should be noted that only data within one hour after the first limonene injection is evaluated as $RO_2$ produced from the further oxidation of organic products are not considered.

Figure 10 shows the modelled NO and $NO_x$ concentrations for the two model runs. In the experiment with medium NO

concentrations on 08 August 2012, $RO_2$ radical concentrations are overestimated by about 50 – 100% by the reference model and modelled NO as well as $NO_x$ concentrations, are 25% lower than measurements. With an additional $RO_2$ loss rate of 0.05 s$^{-1}$ (Table 5), the fraction of $RO_2$ that reacts with NO reduces from 80 – 90% to 45 – 60%. Therefore, the loss of $NO_x$ by the formation of organic nitrates is also reduced, so that the model-measurement agreement for NO and $NO_x$ improves for the first two hours of the experiments.





Both model runs overestimate the $NO_x$ and NO concentrations, when all limonene reacted away after 13:00 UTC (Fig. 5). The measured $NO_x$ concentration remains stable at around 0.6 ppbv throughout the whole experiment after the injection of limonene. However, $NO_x$ concentrations increase at a rate of about 0.15 ppbv h$^{-1}$ in the reference model. The increase is reduced to less than 0.05 ppbv h$^{-1}$ in the model run with the additional $RO_2$ loss during the last two hours of the experiment. The production of $NO_x$ in the model at later times of the experiment can be explained by the production of $NO_2$ from the photolysis of the

first-generation organic nitrates and their oxidation by OH. These effects are more important in the reference model run, when the modelled first-generation organic nitrates are high. To reconcile the difference in $NO_x$ concentrations between the model and measurements, a stronger nitrogen sink is required in the model. This may also suggest that the model underestimates the organic nitrate formation from the reaction of NO with $RO_2$ from the oxidation of product species. Another explanation would be that the lifetime of limonene-derived organic nitrates from OH-oxidation is too short in the model.

In the experiment with low NO concentrations on 01 September 2012, the fraction of $RO_2$ that reacts with NO reduces from about 50% to 16%, if an additional $RO_2$ loss with a rate of 0.06 s$^{-1}$ (Table 5) is applied. In this experiment, a large fraction of $NO_x$ in the model is lost due to the formation of PAN or PAN-like species from acyl peroxy radicals (e.g., CH3CO3 and C822CO3) that are formed in the radical chain reaction of the ozonolysis reaction of limonene (Fig. 1). The additional loss of the initially formed $RO_2$ species competes with the reaction with $NO_2$ and therefore the formation of PAN reduces, if the

additional loss is applied. This effect of reduced $NO_x$ loss in the ozonolysis reaction adds to the effect for a reduced organic nitrate formation discussed for the experiment on 08 September 2012 at medium NO.

Although the reduced $NO_x$ loss significantly improves the model-measurement agreement for the first part of the experiments, if an additional $RO_2$ loss process is included in the model, $NO_x$ concentrations are overestimated by this model at later times, when the chemistry of product species gains in importance. This could be due to neglecting the impact of the subsequent

chemistry of the additional $RO_2$ loss reactions on nitrogen oxide concentrations. The chemistry of nitrogen oxide species in the experiment with low NO concentrations is more complex compared to the experiment with medium NO as a significant fraction of $RO_2$ radicals is produced by the ozonolysis of limonene in addition to the reaction with OH. Further investigation will be required to specifically clarify the impact of the formation of PAN and PAN-liked species from the ozonolysis of limonene (Fig. 1). To our best knowledge, there is no experimental study investigating PAN formation from the oxidation of

limonene.





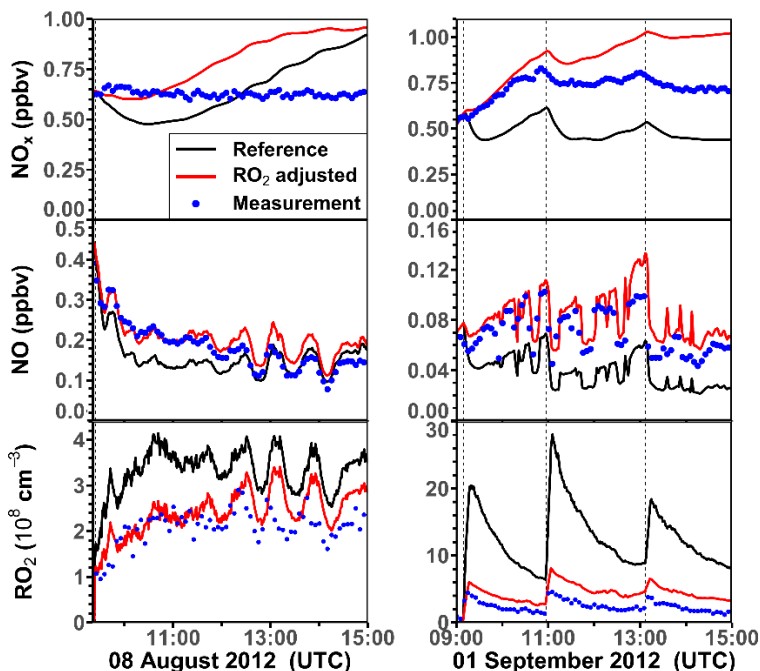

Figure 10. Example of the impact of too high modelled $RO_2$ on the modelled $NO_x$ and NO concentrations in the experiments with medium NO (08 August 2012) and low NO (01 September 2012) concentrations. In both cases, the organic nitrate yield
of 34% from the reaction of NO with first-generation $RO_2$ from the limonene+OH reaction is taken from the analysis in this work.

## 4 Discussion

### 4.1 Sensitivity model runs including additional radical regeneration reactions for $RO_2$ radicals

Additional radical regeneration reactions are further explored by sensitivity model runs. These model runs aim to reduce the discrepancies between measured and modelled OH, $HO_2$ and $RO_2$ concentrations in the reference model. This is achieved by implementing additional reaction pathways:

$$RO_2 + (X) \rightarrow products + HO_2 \qquad\qquad (R1a)$$

$$RO_2 + (X) \rightarrow products + OH \qquad\qquad (R1b)$$

where $RO_2$ are the peroxy radicals distinguished from their production in the reaction of limonene with either OH (limOH-$RO_2$; i.e., LIMAO2, LIMBO2, and LIMCO2, Fig. 1) or $O_3$ (lim$O_3$-$RO_2$; i.e. LIMALAO2, LIMALBO2, and L5O2, Fig. 1), because these peroxy radicals are structurally similar with peroxy radicals from the OH reaction having a β-OH moiety of a 6-carbon ring and limO3-$RO_2$ being acyclic peroxy radicals with a β-oxo, an aldehyde and an isopropenyl group. Therefore, it





is assumed that they have similar reaction pathways. Reactions could involve an unknown reaction partner X like for example
used in Hofzumahaus et al. (2009) or could be unimolecular reactions.

Reaction rate constants for Reaction R1a and R1b ($k_{R1a}$ and $k_{R1b}$) are implemented as pseudo-first order reaction rate constants.
$RO_2$ within the same group (limOH-$RO_2$ or limO$_3$-$RO_2$) are assumed to have the same rate constants. Reaction R1a would lead
to $HO_2$ production and Reaction R1b would lead to OH production. In the sensitivity model runs, the reaction rate constants
are optimized to minimize the model-measurement discrepancies for OH, $HO_2$, $RO_2$ concentrations and OH reactivity. The
sum of $k_{R1a}$ and $k_{R1b}$ must be within the range of the additional $RO_2$ loss rate $k_{add}$ (Section 4.1, Table 5). A missing $HO_2$ source
is found in the reference model for all experiments (Fig. 4 – 7). Assuming that the loss rate of the $HO_2$ radical, which mainly
reacts with NO, is correctly accounted for, an additional $RO_2$ to $HO_2$ conversion (Reaction R1a) is needed to bring
measurement and model results in agreement. In contrast, missing OH is only found in the experiments with low NO as evident
from the analysis of the chemical budget of OH radicals (Section 3.3, Fig. 8). These observations indicate that additional $RO_2$
to OH conversion (Reaction R1b) can only be competitive with other bimolecular reactions for NO mixing ratios of less than
0.05 ppbv which is equivalent to a loss rate of $k < 10^{-2}$ s$^{-1}$.

The model-measurement agreement of radical concentrations is first optimized based on the second half of the ozonolysis
experiment, when CO was added as OH scavenger. In this case, only limO$_3$-$RO_2$ are present, but the conversion to either $HO_2$
(Reaction R1a) or OH (Reaction R1b) cannot be distinguished, because OH rapidly converts to $HO_2$. To achieve agreement
between modelled and measured $HO_2$ concentrations during this part of the ozonolysis experiment, the sum of the additional
loss rates ($k_{R1a} + k_{R1b}$) would need to be $(0.017 \pm 0.008)$ s$^{-1}$. The uncertainty is mainly due to the uncertainty in the measurement
of $HO_2$ concentrations (~20%). The upper limit for the rate constant $k_{R1b}$ for the loss of limO$_3$-$RO_2$ can be estimated from the
first part of the ozonolysis experiment, when no OH scavenger was present. Since 80 to 100% of the observed OH production
can already be explained by OH production from the limonene ozonolysis reaction (Fig. 8d), the rate of $k_{R1b}$ for limO$_3$-$RO_2$
would need to be less than 0.004 s$^{-1}$. This implies that the rate constant $k_{R1b}$ for limOH-$RO_2$ is also less than ~0.004 s$^{-1}$ as about
40% of limonene is oxidized by OH in the ozonolysis experiment without OH scavenger.

The implementation of the Reaction 1a for limO$_3$-$RO_2$ cannot significantly improve the model-measurement discrepancies of
$HO_2$ concentrations in the experiments, when limonene is predominantly oxidized by OH. Also in the ozonolysis experiment,
$HO_2$ concentrations are still underestimated by about 40% during the part of the experiment without OH scavenger. Hence,
the reaction rate $k_{R1a}$ for an additional loss of limOH-$RO_2$ is also optimized to match the measured $HO_2$ concentrations.

Optimization of the reaction rate $k_{R1a}$ for the additional loss of limOH-$RO_2$ for individual experiments results in values that
differ by one order of magnitude. For instance, the optimum rate constant is $(0.006 \pm 0.003)$ s$^{-1}$ in the ozonolysis experiment
without OH scavenger, but it is $(0.05 \pm 0.03)$ s$^{-1}$ in the experiment with medium NO concentrations on 08 August 2012. These
optimizes rate constants are consistent with the values of the loss rate $k_{add}$ (Table 5), with the rate required in the ozonolysis
experiment having a slower rate while the rate required in the experiment with medium NO having a faster rate. The exact





reason for such large differences is not clear, but could be related to the higher temperature ($16 - 27°C$) in the photo-oxidation experiments, when the chamber air was exposed to sunlight. The average value of the rate constant for the conversion from $RO_2$ to $HO_2$ for the experiments in this work is 0.03 $s^{-1}$. This value is applied to all sensitivity model runs in the following to illustrate its impacts on modelled $RO_2$, $HO_2$ concentrations and OH reactivity (Table 6).

A summary of all reactions included for the sensitivity run is available in Table 6. Figure 11 shows the increase in the OH production rate in the sensitivity model runs that include the conversion of $limO_3$-$RO_2$ to OH at a rate of 0.004 $s^{-1}$. The total OH production rate increases by about 0.2 ppbv $h^{-1}$ in both experiments corresponding to a 5% and 10% increase, respectively. This reduces the imbalance between OH production and destruction rates in the experiment with low NO by about 20% without significantly impacting the balance in the ozonolysis experiment. This demonstrates that the additional OH production from

the conversion of first-generation $RO_2$ from OH- or $O_3$- oxidation of limonene to OH is not sufficient to fully close the gap between OH production and destruction rates in the experiment with low NO, for which the discrepancy is largest among all experiments in this work.

Figure 12 and Figure S10 show radical concentrations and OH reactivity obtained in the reference and sensitivity model runs. In the sensitivity model run, the model-measurement agreement for $RO_2$ and $HO_2$ concentrations improves compared to the

reference model run as can be expected from the adjustment of the reaction rate constant. In the experiment with low NO concentrations, however, an optimal agreement of both, $RO_2$ and $HO_2$ concentrations, cannot be simultaneously achieved. This suggests that some fraction of the additional $RO_2$ loss pathway may not regenerate $HO_2$ or OH radicals.

In the sensitivity model run the overestimation of the OH reactivity is reduced even without introducing additional loss pathways of oxidised products as implemented in the constrained model run (Section 2.4), because the production of organic

peroxides is reduced due to the competition with the additional $RO_2$ loss reaction. For example, the percentage of $RO_2$ reacting with $HO_2$ reduces from about 50% to 25% and from 90% to 30% in the experiment with low NO and in the ozonolysis experiment, respectively. In the sensitivity model, no closed-shell products are produced from the additional $RO_2$ loss reaction. The good model-measurement agreement of the OH reactivity suggests that organic products from these reactions are not reactive or they are rapidly lost for example to the chamber wall. Therefore, no further conclusions, about the type of products

formed from Reaction R1 can be drawn from these experiments.






**Table 6.** Modification of chemical reactions implemented in the sensitivity simulations runs.

| Reaction | Reaction rate constant | Comment |
|---|---|---|
| LIMALAO2 → OH | 0.003 s$^{-1}$ | Illustrate the impact of the additional OH source on the OH budget in the ozonolysis experiment, when there is no OH scavenger. |
| LIMALBO2 → OH | 0.003 s$^{-1}$ | |
| L5O2 → OH | 0.003 s$^{-1}$ | |
| LIMALAO2 → HO2 | 0.014 s$^{-1}$ | Derived from the optimisation of HO$_2$ model-measurement agreement in the ozonolysis experiment when there is OH scavenger. Assuming $k_{R1b}$ for limO$_3$-RO$_2$ is 0.003 s$^{-1}$ |
| LIMALBO2 → HO2 | 0.014 s$^{-1}$ | |
| L5O2 → HO2 | 0.014 s$^{-1}$ | |
| LIMAO2 → HO2 | 0.030 s$^{-1}$ | Mean value of the rate derived from the optimisation of HO$_2$ model-measurement agreement in the experiments with low NO, medium NO and the ozonolysis experiment. |
| LIMBO2 → HO2 | 0.030 s$^{-1}$ | |
| LIMCO2 → HO2 | 0.030 s$^{-1}$ | |

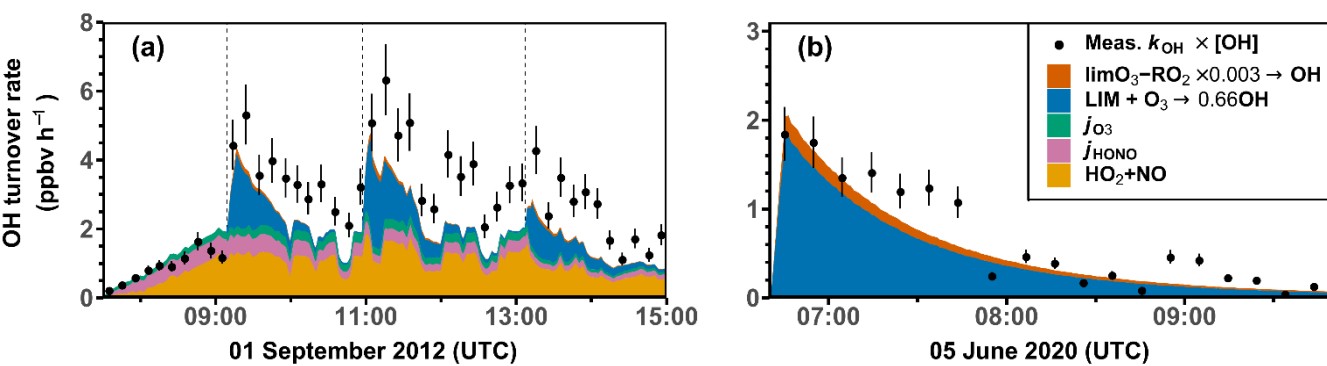


**Figure 11.** Measured 10-minutes-mean total OH production rate compared with the OH production rate from the main measured OH sources for (a) the experiment with low NO concentration on 01 September 2012 and (b) the ozonolysis experiment on 05 June 2020 for the sensitivity model run that includes additional OH production from the reaction of RO$_2$ from limonene ozonolysis and HO$_2$ (Reaction R2c).





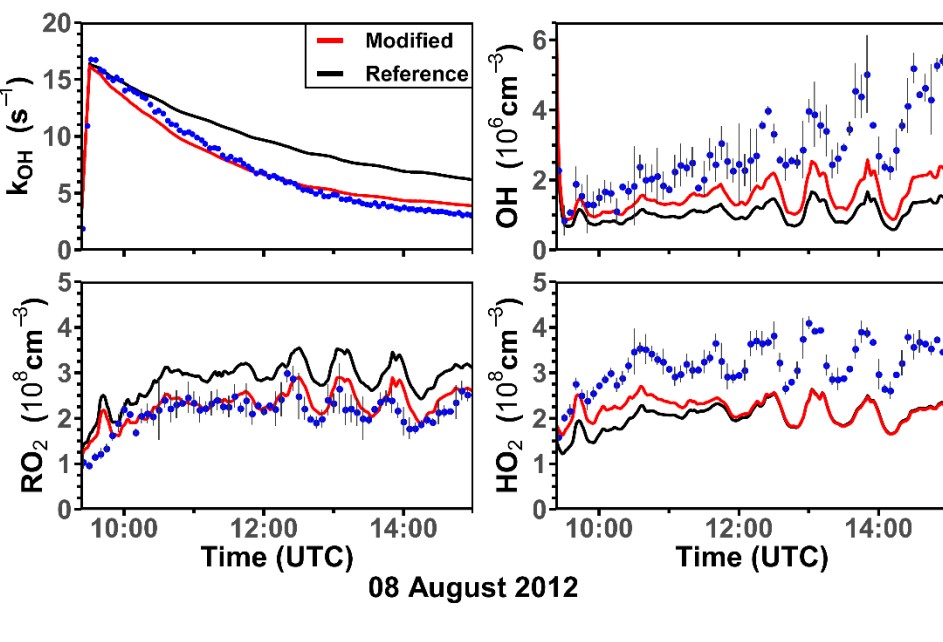

**08 August 2012**


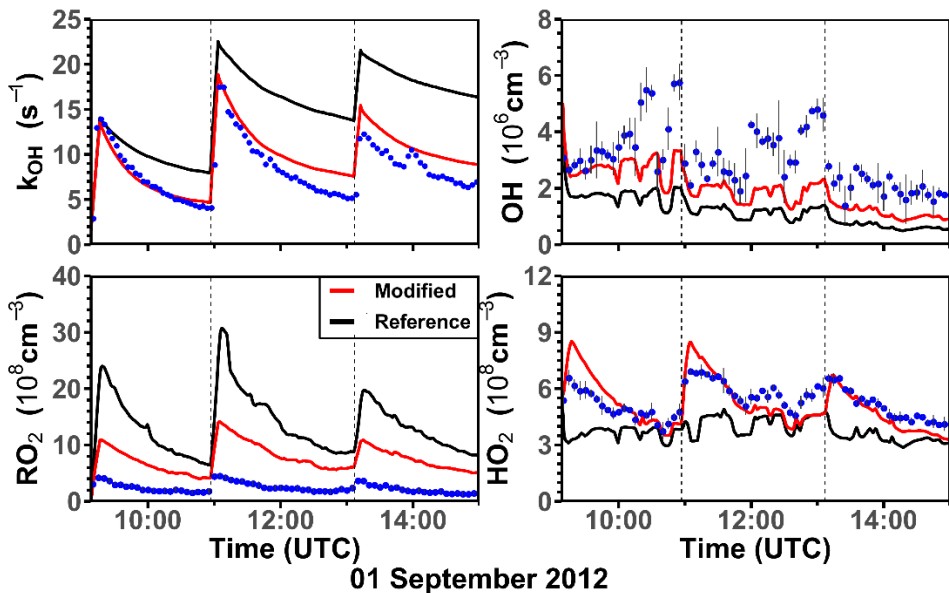

**01 September 2012**





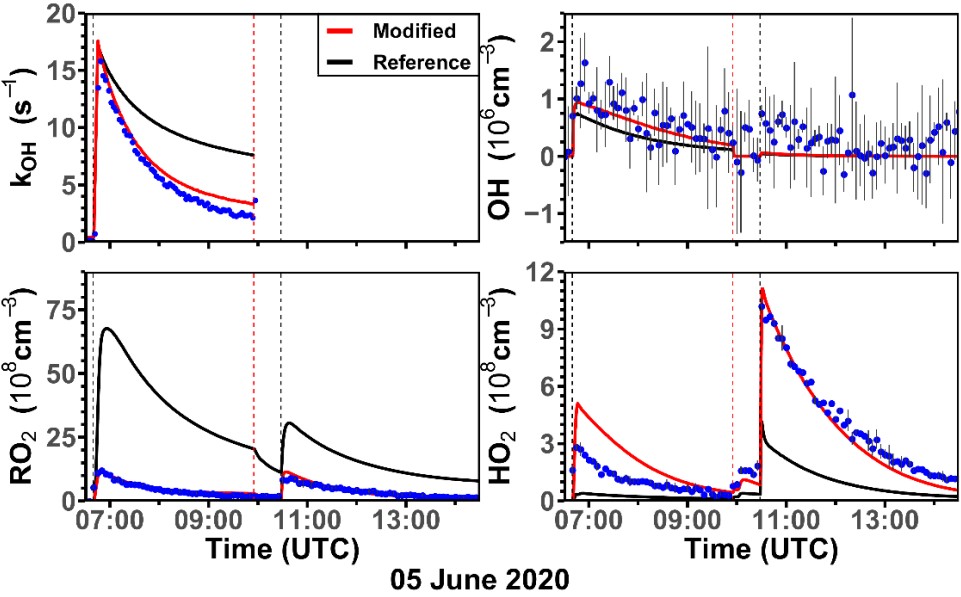

**Figure 12.** Modelled and measured OH reactivity, HO₂, RO₂, and OH concentrations for the experiments with medium NO
(top) and low NO (middle) concentrations, and for the ozonolysis experiment (bottom). Results from the reference model
(reference) and the sensitivity model run (modified) that includes additional RO₂ loss processes producing OH and HO₂ (Table
6) are compared.

## 4.2 Possible additional OH sources from RO₂ reactions

Possible underlying mechanisms of the additional RO₂ loss and the production of OH are discussed in the following section.
It should be noted that the additional OH source discussed here is referring to the slow additional OH source ($k_{R1b} < 0.003$ s⁻¹)
that could only slightly reduce the discrepancy between the OH production and destruction rates (Fig. 11). The larger
discrepancy in the chemical OH budget that is observed in the experiment with low NO concentrations (Fig. 11a) compared
to the other experiments cannot be explained by the additional conversion of RO₂ to OH and, hence, it is not further discussed
here. The rate of the additional RO₂ to OH conversion, $k_{R1b}$, is much slower than the total loss rate of RO₂ due to bi-molecular
reactions ($k_{bi} > 0.01$ s⁻¹) or the additional RO₂ to HO₂ conversion ($k_{R1a} \sim 0.017$ s⁻¹). Its contribution to the production rate of
OH would also be small in all experiments (< 10% of the total OH production rate). Therefore, there is no strong evidence for
an additional conversion of RO₂ to OH for experimental conditions in this study.



### 4.2.1 RO₂ isomerisation reactions

Isomerisation of $RO_2$ could lead to the production of OH, which is shown in the OH-oxidation of methacrolein (e.g., Crounse et al., 2012) and isoprene (e.g., Novelli et al., 2020). Isomerisation reactions for $RO_2$ from the limonene oxidation by OH and $O_3$ are investigated by Møller et al. (2020) and Chen et al. (2021), respectively. Of the peroxy radical LIMAO2, LIMBO2 and LIMCO2 that are produced from the limonene oxidation by OH, it was calculated that LIMAO2 and LIMBO2 could undergo -OH H-shift with a rate constant of $10^{-3}$ to $10^{-4}$ $s^{-1}$. LIMCO2 could undergo isomerisation reactions that are competitive with

bimolecular reactions for atmospheric conditions, which include a cyclization reaction with a rate constant of 0.2 to 0.8 $s^{-1}$ and an allylic-H shift reaction with a rate constant of 0.1 to 1.7 $s^{-1}$ (Fig. 13). On the other hand, peroxy radicals LIMALAO2, LIMALBO2, and L5O2 that are produced from the ozonolysis of limonene could all undergo much faster isomerisation reactions ($k > 0.5$ $s^{-1}$) such as aldehydic H-shift and cyclisation reactions (Fig. 14) due to the loss of steric hindrance after the ring-opening ozonolysis reaction.

Subsequent reaction steps after the first isomerisation reaction were not investigated in the work by Møller et al. (2020) and by Chen et al. (2021). Therefore, potential OH production from subsequent reactions are estimated with available SAR (e.g., Vereecken and Peeters, 2009; Vereecken and Nozière, 2020; Vereecken et al., 2021).

The subsequent reaction of β-OOH substituted alkoxy radicals, LIMA_15shift_O and LIMB_15shift_O, which are products of the 1,5 -OH H-shift reactions of LIMAO2 and LIMBO2 respectively, is a ring-breaking decomposition reaction that

produces an OH radical. For the cyclization product from the isomerisation of LIMCO2, LIMC_6cyc_O2, it may either undergo bimolecular reactions or isomerisation by abstracting the α-H atom of the hydroxyl group (Fig. 13). Isomerisation of LIMC_6cyc_O2 could result in an aldehyde together with an $HO_2$ radical. However, it is uncertain, if the isomerisation reaction of LIMC_6cyc_O2 can compete with other reaction channels, as the SAR does not apply to bicyclic compounds. The reaction between LIMC_6cyc_O2 and NO results in a bicyclic alkoxy radical (LIMC_6cyc_O), which then dissociates to a cyclic

intermediate, analogous to the bicyclic alkoxy radical produced from the OH-oxidation of β-pinene (Vereecken and Peeters, 2004). The dissociation of LIMC_6cyc_O may eventually produce an $HO_2$ radical, which could not explain the production of OH at zero NO condition.

For $limO_3$-$RO_2$, their subsequent reactions after the first isomerisation reaction are also expected to be very fast ($k > 0.1$ $s^{-1}$) because of the presence of either an aldehyde group or a C=C double bond. However, allylic H-shift, cyclisation, and aldehydic

H-shift reactions do not affect the $RO_2$ and OH concentrations as the resulting alkyl radicals do not decompose into a closed-shell product and an OH radical. Instead, an oxygen molecule rapidly adds to the alkyl radical site to form a highly oxidised $RO_2$.

One of the possible additional OH sources from $RO_2$ isomerisation reactions could be from the H-abstraction of the α-OOH group (Bianchi et al., 2019). In this case, the α-OOH substituted alkyl radical could quickly decompose into a carbonyl product

and an OH radical. Because of the fast first and second step of the isomerisation reaction, it is expected that $limO3$-$RO_2$ could



form $RO_2$ with a hydroperoxide group that allows them to undergo α-OOH H-shift reaction (e.g., LIMALA_17alde_O2, LIMALB_19alde_O2, L5_5cyc_17alde_O2 (Fig. 14)). The SAR-estimated α-OOH H-abstraction rate is about $10^{-3}$ to $10^{-2}$ $s^{-1}$ (Vereecken and Nozière, 2020), which is slightly faster than the additional $RO_2$ loss rate $k_{R1b}$ (< 0.003 $s^{-1}$) applied in the sensitivity run. It should be noted that the SAR does not consider the impacts of neighbouring functional groups on the α-OOH
H-shift rate. All $limO_3$-$RO_2$ have a β-oxo group near the radical site, which could increase the H-abstraction rate near the β-oxo group by about one order of magnitude (e.g., Crounse et al., 2013), including the α-OOH H-abstraction. However, $RO_2$ derived from the isomerisation of $limO_3$-$RO_2$ still have other possible isomerisation reaction pathways that are even more competitive ($k > 1$ $s^{-1}$) than the α-OOH H-shift. One examples is the H-shift reaction from the hydroperoxyl group to another peroxy group (Vereecken and Nozière 2020). This complicates the chemistry of these highly oxidized peroxy radicals and it
is unclear, whether they eventually undergo an α-OOH H-abstraction.







**Figure 13.** Unimolecular reaction pathways of $RO_2$ radicals from the reaction of limonene with OH reported in Møller et al. (2020) and subsequent anticipated reaction pathways. Reaction rate constants denoted with $k_{SAR}$ are directly taken from SAR for peroxy radicals (Vereecken and Nozière, 2020). The unimolecular reaction rate for 1,8 α-OH H-shift is not available.





**Figure 14.** Unimolecular reaction pathways for the major limonene-ozonolysis-derived RO$_2$ radicals. Rates taken from calculations in Chen et al. (2021), in which stereoisomers are considered and therefore a range of values are given. Other rates ($k_{SAR}$) are based on SAR in Vereecken and Nozière (2020) and Vereecken et al. (2021). It should be noted that their SAR does not address the effect of steric hindrance (in the case of a cyclic compound) or multiple functional groups near the abstracted H-atom on the H-shift rate constant.





### 4.2.2 Reactions between $RO_2$ and $HO_2$

Another possible $RO_2$ loss reaction that leads to OH production and is not considered in the reference model is the reaction of β-oxo-substituted $RO_2$ with $HO_2$. Based on the work by Jenkin et al. (2019), $RO_2+HO_2$ reactions can lead to different products depending on the functional groups nearby the peroxy group of the $RO_2$ radical:

$\quad RO_2 + HO_2 \rightarrow ROOH + O_2$ (R2a)

$RO_2 + HO_2 \rightarrow ROH + O_3$ (R2b)

$RO_2 + HO_2 \rightarrow RO + OH + O_2$ (R2c)

$RO_2 + HO_2 \rightarrow R_{-H}=O + H_2O + O_2$ (R2d)

$RO_2 + HO_2 \rightarrow R_{-H}=O + OH + HO_2$ (R2e)

It has been shown in experimental studies investigating simple β-oxo substituted $RO_2$ that they can form an alkoxy radical together with an OH radical (Reaction R2c) instead of a hydroperoxide compound (Reaction 2a) (Jenkin et al., 2007; Hasson et al., 2012; Praske et al., 2015). With the branching ratios that are taken from the SAR, which is derived using simple β-oxo substituted $RO_2$, the OH yield of the $RO_2+HO_2$ reaction for $limO_3$-$RO_2$ is about 30% (Jenkin et al., 2019). This value is similar to the OH production rate in the sensitivity runs using an additional $RO_2$ loss rate of $k_{R1b} = 0.003$ s$^{-1}$. However, theoretical

investigation by Iyer et al. (2018) suggests that the OH yield of the $RO_2+HO_2$ reaction may not be high enough ($< 1\%$) based on the energy barriers that were calculated in that study for $limO_3$-$RO_2$. Currently, there is no laboratory study on the OH yield from the $RO_2+HO_2$ reaction for large $RO_2$ and therefore the OH yield of the $RO_2+HO_2$ for large $RO_2$ is highly uncertain.

### 4.3 Possible additional $HO_2$ sources from $RO_2$ reactions

**4.3.1 $RO_2$ isomerisation reactions**

Similar to the additional OH source, $HO_2$ can also be produced from the isomerisation reaction of $RO_2$ radicals (e.g., Crounse et al., 2012; Peeters et al., 2014). Again, the possibility of additional conversion from $RO_2$ to $HO_2$ by isomerisation is investigated using the isomerisation pathways calculated for $RO_2$ derived from limonene oxidation and SARs (Vereecken and Peeters, 2009; Møller et al., 2020; Vereecken and Nozière, 2020; Chen et al., 2021; Vereecken et al., 2021).

Peroxy radicals LIMAO2 and LIMBO2 that are produced from the OH-oxidation of limonene can undergo a slow ($k < 10^{-3}$ s$^{-1}$) -OH H-shift reaction that is one order of magnitude slower than the rate of the $RO_2$ to $HO_2$ conversion applied in the sensitivity run (Fig. 13). In addition, the production of $HO_2$ through the H-abstraction by $O_2$ of the alkoxy radical LIMA_15shift_O is not as favourable as the ring-cleavage alkoxy dissociation that eventually produces an OH radical





(Vereecken and Peeters, 2009). Therefore, even with the potential production of $HO_2$ through the isomerisation of $RO_2$ derived from the isomerisation of LIMCO2 (e.g., 1,8 α-OH H-shift of LIMC_6cyc_O2), the production of $HO_2$ is limited by the 37% yield of LIMCO2 from the oxidation of limonene by OH. In addition, the rate constant of the H-shift reaction for the intermediate radicals derived from LIMCO2 (e.g., LIMC_6cyc_O2, LIMC_16allylic_O2, LIMC_15allylic_O2) have a high uncertainty as it is assumed that the SAR for acyclic compounds can be applied for cyclic $RO_2$. For these reasons, isomerisation reactions for limOH-$RO_2$ are unlikely the reason for the additional $HO_2$ production and $RO_2$ loss required to match observed radical concentration measurements.

For limO$_3$-$RO_2$, reaction rate constants of the first two of the isomerisation reaction are about one to two orders of magnitude faster than the $RO_2$ to $HO_2$ conversion rate used in the sensitivity run (Fig. 14). As discussed in Section 4.2.1, one of the possible $RO_2$ loss mechanisms through isomerisation reaction is a α-OOH H-abstraction reaction. Although the value of the rate constant of the α-OOH H-abstraction reaction derived from SAR is on the same magnitude ($\sim 10^{-2}$ s$^{-1}$) as the $RO_2$ to $HO_2$ conversion rate applied in the sensitivity run, abstraction of the hydrogen with an α-OOH group would lead to the production of an OH radical rather than a $HO_2$ radical. Therefore, isomerisation reactions of limO$_3$-$RO_2$ can also not explain the missing $RO_2$ to $HO_2$ conversion resulting from observations in the experiments in this work.

### 4.3.2 Reaction rate of the RO$_2$ recombination reaction

Apart from isomerisation, $HO_2$ could also be produced from the dissociation of alkoxy radicals derived from $RO_2$ from the reaction of limonene with OH. Alkoxy radicals could be produced from the recombination reaction of $RO_2$ radicals in addition to the reaction of $RO_2$ with NO:

$$RO_2 + R'O_2 \rightarrow RO + R'O + O_2 \tag{R3a}$$

$$RO_2 + R'O_2 \rightarrow ROH + R'(=O) + O_2 \tag{R3b}$$

$$RO_2 + R'O_2 \rightarrow R(=O) + R'OH + O_2 \tag{R3c}$$

$$RO_2 + R'O_2 \rightarrow ROOR' + O_2 \tag{R3d}$$

The current knowledge about the branching ratio between Reaction R3a to R3d, as well as the $RO_2$ self- or cross-reaction rate constants is limited especially for complex $RO_2$ derived from monoterpenes. There are no specific investigations for $RO_2$ from limonene. Reaction rate constants implemented in the MCM model are based on estimated cross-reaction rates between $RO_2$ and methyl peroxy radicals ($CH_3O_2$), (Jenkin et al., 1997, 2019). The reaction rate constants of the $RO_2$ recombination reactions, $k_{RO2+RO2}$, for limonene-derived radicals are between $10^{-12}$ to $10^{-13}$ s$^{-1}$ in the MCM consistent with results for $RO_2$ from methyl cyclohexene, which contain a tri-substituted endocyclic double bond like limonene (Boyd et al., 2003).





However, the reaction rate constant $k_{RO2+RO2}$ could be higher, if the cross-reaction partners are other large limonene-derived radicals rather than $CH_3O_2$. For example, Berndt et al., (2018) investigated the self-reaction rate constants for $RO_2$ derived from the reaction of α-pinene with OH after they undergo two steps of unimolecular reactions (i.e., $C_{10}H_{16}OH(O_2)_2$-$O_2$). They found that values range between 1 and $4\times10^{-11}$ cm$^3$ s$^{-1}$ in this case. However, it should be noted that these high rates are derived from the production rate of peroxide products (ROOR, Reaction R3d), rather than the loss rate of $RO_2$.

Using the values of the reaction rate constants $k_{RO2+RO2}$ for $RO_2$ from limonene oxidation from the MCM, the upper limit of the $RO_2$ loss rate due to $RO_2$-$RO_2$ reactions is about $10^{-3}$ s$^{-1}$ in the ozonolysis experiment and experiments with low NO, and $2\times10^{-4}$ s$^{-1}$ in the experiments with medium NO. From the additional loss rate (~$10^{-2}$ s$^{-1}$; Table 5) determined from the chemical budget analysis for $RO_2$, the value of the reaction rate constant for $RO_2$-$RO_2$ reaction that would be required to explain the observations ($k'_{RO2+RO2}$) can be calculated. This results in values of $k'_{RO2+RO2}$ that are about $3\times10^{-10}$, $1\times10^{-11}$, $3\times10^{-11}$ cm$^3$ s$^{-1}$ in the medium NO, low NO and ozonolysis experiments, respectively. The uncertainties of the rates are about 50 – 60%, which are derived from the error propagation of $RO_2$ concentrations and optimal rate constants in the sensitivity model run (Table 6). It should be noted that these values are collective loss rates of all first-generation $RO_2$ species from limonene oxidation before the formation of closed-shell products, including highly-oxidized $RO_2$ produced from potential auto-oxidation reactions.

The values of the reaction rate constant $k'_{RO2+RO2}$ found in the low NO experiment and ozonolysis experiment are in the same order of magnitude ($10^{-11}$ to $10^{-10}$ cm$^3$ s$^{-1}$) as values reported by Berndt et al., (2018) for $RO_2$ from α-pinene oxidation. Berndt et al., (2018) also showed that the reaction rate constant for the $RO_2$-$RO_2$ self-reaction increases, when the $RO_2$ becomes more oxidised. This hints the importance of $RO_2$ recombination reactions for $RO_2$ derived from limonene oxidation could be higher than previously thought, because of the rapid isomerisation reaction of these radicals (Møller et al. (2020) and Chen et al. (2021); Section 4.2.1).

In the experiment with low NO concentrations, the additional loss rate for $RO_2$ radicals that is required to explain measured $RO_2$ concentrations ($k \sim 0.06$ s$^{-1}$, Table 5) is higher than the rate of the additional $RO_2$ to $HO_2$ conversion required to explain measured $HO_2$ concentration ($k \sim 0.006 – 0.02$ s$^{-1}$). This would be consistent with a faster reaction rate constant for the $RO_2$-$RO_2$ reaction, because only a fraction of $RO_2$-$RO_2$ reaction would lead to the formation of alkoxy and therefore $HO_2$ radicals (Reaction 3).

However, it would be unclear, why the reaction constant $k'_{RO2+RO2}$ required in the experiment with medium NO mixing ratio would be higher compared to other experiments. It is also worth noting that the decomposition of alkoxy radicals produced from $RO_2$ from the ozonolysis of limonene leads to the production of peroxy radicals which do not lead to the production of $HO_2$ in most of the $RO_2$-$RO_2$ reaction chain (Fig. 1). Therefore the missing production of $HO_2$ in the ozonolysis experiment cannot be explained by a higher than previously thought reaction rate constant of the $RO_2$-$RO_2$ reaction.





## 5 Conclusions

The photooxidation of limonene by OH and O₃ was investigated in experiments for zero, low (~0.1 ppbv), medium (~0.4 ppbv) and high (~ 1 ppbv) NO levels in the atmospheric simulation chamber SAPHIR. The experiments were conducted with limonene mixing ratios of 4 – 10 ppbv and O₃ mixing ratios ranging from 0 – 50 ppbv.

The analysis of measured radical concentrations in the experiments revealed that current knowledge about the limonene oxidation as implemented in the Master Chemical Mechanism cannot explain observed values specifically concerning radical

regeneration. Observed OH and HO₂ concentrations were a factor of 2 to 3 higher than predicted by model calculations, whereas measured RO₂ concentrations were at least 50% lower than modelled values. The following processes in the limonene mechanism impacting radical concentrations could be identified that are not appropriately described:

- The loss rate of OH radicals is too high in the model as seen in higher-than-observed OH reactivity values in the model. Although it cannot be excluded that chamber wall losses reduced the concentration of organic oxidation

products, this hints that the reactivity of products species with respect to their reaction with OH is low or other products other than currently thought are produced for example by the competition of unaccounted radical reaction pathways.

- The yield of organic nitrates from the reaction of RO₂ radicals formed in the initial reaction of OH with limonene is found to be (34±5)%, which agrees with the measurements by Rollins et al. (2010), but which is about 10% higher

than calculated from structure-activity relationships. The higher yield of organic nitrates reduces the efficiency of the radical regeneration in the limonene mechanism.

- Formaldehyde is expected to be formed from the reaction chain after the addition of OH to the terminal C=C double bond, if RO₂ radicals react with NO, so that the formaldehyde yield would be similar to the yield of that RO₂ species (37%). The low formaldehyde yield of (13±3)% in the experiments with medium NO concentrations suggests that

there is an unaccounted RO₂ loss reaction not producing formaldehyde that is competitive at 200 pptv NO.

- OH production and destruction rates are balanced in most of the experiments, if measured OH reactivity and measured HO₂ concentrations are used for calculating reaction rates. This demonstrates that measured values are consistent and confirm the shortcomings of the limonene mechanism with regards to describing the HO₂ production and OH reactivity.

- An unaccounted RO₂ loss process with a rate of 0.02 to 0.06 s⁻¹ is required to balance the RO₂ production rate from the reaction of OH with organic compounds. Formation of HO₂ with a rate of 0.03 s⁻¹ and 0.017 s⁻¹ from an additional reaction of the RO₂ from the reaction of limonene with OH and O₃, respectively, can explain part of the model-measurement discrepancies for HO₂.

- An unaccounted RO₂ loss process for RO₂ from the ozonolysis of limonene that is competitive against the reaction

with NO prevents the formation of NOₓ reservoir species PAN and PAN-like species as suggested in the MCM model.





The observed $NO_x$ concentrations do not exhibit a distinct temporal behaviour that would be expected neither from the rapid loss of $NO_x$ species at the beginning of limonene oxidation when $RO_2$ derived from limonene oxidation reacts with $NO_x$, and $NO_x$ reformation from the thermal decomposition of PAN species at the later times of the experiments when limonene has reacted away.

Overall, the results of the experiments clearly demonstrate that loss reactions of $RO_2$ from the oxidation of limonene are not well understood. Unaccounted $RO_2$ reactions lead to an enhanced radical regeneration. Organic products likely are less reactive than products that are currently thought to be formed. Time series of measured radical concentrations indicate that their further oxidation reactions need to be investigated to explain observed values at later times of the experiments, when limonene had reacted away. The formation of $NO_x$ reservoir species (PAN / PAN-like species) is lower than expected. However, this is partly

counter-acted by a high yield of $(34\pm5)\%$ of organic nitrates. Oxidation products are also likely to have a high organic nitrates yield as indicated by the low measured $NO_x$ concentrations that would otherwise be expected to continuously increase over the course of the experiment due to the continuous emission of HONO by the chamber film.

Rates of $RO_2$ isomerisation reactions proposed by Møller et al. (2020) and Chen et al. (2021) are too low or too fast ($k \sim 10^{-4}$ to $10^{-3}$ and 1 to $10^{2}$ $s^{-1}$) to explain observed radical concentrations and expected products are not consistent. A possible

explanation could be that the reaction rate constant $k_{RO2+RO2}$ of $RO_2$ recombination reactions for $RO_2$ from limonene oxidation is higher than the reaction rate constants that are calculated by SAR implemented in the MCM. For example, experiments for $RO_2$ from $\alpha$-pinene oxidation by Berndt et al. (2018) show that the rate constant can be one to two orders of magnitude faster than implemented in the MCM model for $\alpha$-pinene ($10^{-13}$ to $10^{-12}$ $cm^3$ $s^{-1}$). These values are consistent with the additional loss rate required to explain radical concentrations in the experiments with limonene in this work. However, the importance of the

alkoxy pathway of the $RO_2+RO_2$ reaction for large monoterpene-derived $RO_2$ is still unclear and needs further investigation.

*Data availability*: Data from the experiments in the SAPHIR chamber used in this work are available on the EUROCHAMP data home page. 08 August 2012: https://doi.org/10.25326/JNMN-YC22 (Fuchs et al., 2021b); 10 August 2012: https://doi.org/10.25326/2PGS-FP66 (Fuchs et al., 2021c); 01 September 2012: https://doi.org/10.25326/77N2-ZK22 (Fuchs

et al., 2021a); 03 August 2015: https://doi.org/10.25326/BP56-WP95 (Bohn et al., 2021c), 04 July 2019: https://doi.org/10.25326/C4SW-TP73 (Bohn et al., 2021b); 05 June 2020: https://doi.org/10.25326/7ZKX-347 (Bohn et al., 2021a).

*Supplement*. The supplement related to this article is available online at:






*Author contributions*. JYSP, AN and HF wrote the manuscript. MK, AN and HF designed and led the experiments in the chamber. BB (radiation), RT, AL, IHA and RW (organic compounds), XL (HONO), FR (ozone, nitrogen oxides), HPD and PC (radicals), SN (OH reactivity), CC and AN (radicals and OH reactivity) were responsible for measurements used in this work. All co-authors commented and discussed the manuscript and contributed to the writing of the manuscript.


*Competing interests*. Astrid Kiendler-Scharr and Andreas Hofzumahaus are editors of ACP.

*Acknowledgements*. The authors thank Luc Vereecken for the discussion on the chemical mechanism.

*Financial support*. This project has received funding from the European Research Council (ERC) under the European
Union's Horizon 2020 research and innovation programme (SARLEP grant agreement No. 681529) and from the European Commission (EC) under the European Union's Horizon 2020 research and innovation programme (Eurochamp 2020 grant agreement No. 730997).

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
