# Peer review of "Investigation of the limonene photooxidation by OH at different NO concentrations in the atmospheric simulation chamber SAPHIR"

_Atmospheric Chemistry and Physics, 2022_

## Author Response (AR1)

**Responses to the comments from Referee 1:**

We thank the reviewer for the useful comments improving our manuscript.

**Line 58: no comma needed here**

Authors response:

Thanks for the correction, the comma in line 59 is now removed.

**Line 75: should 'drawing' be 'withdrawing' ?**

Authors response:

To avoid confusion, we rephrase the sentences without using 'drawing' in Line 76 – 79: "H-shift reactions are very slow ($k < 10^{-3}$ s$^{-1}$ at 298 K) in an aliphatic peroxy radical without an oxygenated function group (e.g., carbonyl, hydroxyl, alkoxy) attached to the carbon atom, from which the hydrogen is abstracted (Otkjær et al., 2018; Praske et al., 2019). Therefore, H-shift reactions typically cannot compete with bimolecular reactions under atmospheric conditions ($k_{bi} \sim 10^{-2}$ s$^{-1}$, for 50 pptv of NO and $5\times10^8$ cm$^{-3}$ of HO$_2$)."

**Line 207/8: Could the authors specify that these are (I presume) assumed to be from processes occurring on the chamber film surface?**

Authors response:

Yes, HONO and VOCs are presumably formed from the reactions on the chamber wall surface. We added in Line 217: "…presumably from chamber wall reactions."

**Line 205: This could do with a little more discussion highlighting that the majority of previous experiments will have been done at low humidity. The conditions employed here are clearly more relevant to the atmosphere, but do you have thoughts on whether this may affect the major oxidation pathways. Also, does the high humidity have any negative effect on the instrumentation?**

Authors response:

Statements that mentioned the experimental conditions of the previous experiments are added in the introduction (line 92 – 95): "Radiation and relative humidity during the experiments were also relevant to the conditions that are typically found in the atmosphere, which was an improvement compared to previous experiments that typically used artificial light sources or were conducted under very dry conditions (e.g., Larsen et al. (2001) and Librando and Tringali (2005))."

As far as we know, stabilised Criegee Intermediates (sCI) that are produced from the ozonolysis of limonene can react with water molecules. However, the sCI yield of limonene ozonolysis is about 32% (Cox et al., 2020) and it is estimated that less than 50% of the sCI reacts with water under atmospheric conditions (Vereecken et al. 2017). Therefore, we think that the variation of humidity does not affect the major oxidation pathways of limonene ozonolysis.

In addition, hydroperoxyl radical ($HO_2$) can form an $HO_2$-$H_2O$ complex with a water molecule. The complexation reaction can speed up the self-reaction of $HO_2$ to form hydrogen peroxide. Water vapour can also affect the size distribution of aerosol particles which could change the rate of the uptake reactions. However, the heterogeneous reactions are assumed to be not important in this study.

Humidity can affect the measurement of OH and $NO_x$ concentrations. For example, water vapour can interfere with the measurements of OH using laser-induced fluorescence (LIF) by quenching the OH radical (Holland et al., 1995). However, these instruments are calibrated to minimize the interference in the measurements. Therefore, we do not think that the presence of humidity has a large impact on the experimental results.

**Table 2: Is this the correct NO concentration for the high NO experiment? Is it lower than expected because it's an average? It then doesn't agree with the value used for NO in Table 3.**

Authors response:

Radicals concentrations and OH reactivities are analysed only in part of the experiment with high NO concentrations. Values in Table 2 give concentrations for that period. The NO concentration in Table 3 is from the period after the first limonene injection, when the HCHO yield is calculated. In order to explain this we added in Line 232-238: "When most of the limonene was consumed within two hours after the first injection, an additional injection of 10 ppbv of limonene was done. In this work, the HCHO yield is only analysed based on the measurement before the second limonene injection (Section 3.1.1), because of the potential secondary production of HCHO from the oxidation of secondary products. The radical concentrations and OH reactivities are only analysed after the second limonene injection (Section 3.2.3), because radical measurements failed during the first part of the experiment. A large fraction of NO was already titrated by ozone after the second limonene injection." The caption for Table 2 is also changed accordingly: "Experimental conditions during the time period when the radical budget of limonene oxidation experiments is analysed …"

**Line 361: Highlight that this is as would be expected based on your OH+limonene experiments at low NO, which have a similar OH yield to the ozonolysis experiment.**

Author response:

We assume that the reviewer means HCHO yield. We added in Line 376 – 380: "The HCHO yield derived from the ozonolysis without the presence of the OH scavenger is similar to the HCHO yield in the experiments with low NO concentrations. This is excepted because of the very low NO concentrations and the similar fraction of limonene that reacted with OH or $O_3$ in both experiments."

**Line 368: I'm not sure that this is worth noting without some further explanation of what you mean. Which experiments of Gong et al. does this refer to? All of them? This fact could mean different things based on the experiment. Is it because $O_3$is so high that the $O_3$+ limonene reaction is still dominant over the OH reaction? Or because, as in your experiments, OH is reacting with limonene, but, at low NO, the HCHO yield is similar to the ozonolysis?**

Authors response:

The experiment that we are referring to is the experiment with high limonene:ozone concentration ratio (1:2). This is because HCHO could be produced quickly from the ozonolysis reaction of the secondary products if the ozone concentration is very high. In our analysis, we try to exclude the potential HCHO production from the ozonolysis of secondary products by only considering the measurements when less than 40% of the injected limonene was reacted. In this case, using the reported HCHO yield from Gong et al. with high limonene:ozone concentration ratio to compare with our reported yield is more appropriate. Gong et al. also investigated the effects of humidity and OH scavenger on the HCHO yield. We compared our HCHO yield with the yield reported from the experiments that were conducted under similar relative humidity (30 – 50%) in Gong et al. to see whether the effects of OH scavenger on HCHO yield are consistent in the two studies. The result from Gong et al suggests that OH scavenger does not affect HCHO yield when limomene:ozone concentration is high, which is consistent with our findings. Additional description is now included in Line 388 – 393: "The effects of humidity and presence of an OH scavenger on the HCHO yield were also investigated in Gong et al. (2018). In their experiments, the HCHO yield increases strongly with increasing humidity and in the absence of an OH scavenger when the limonene:$O_3$ ratio was very low (1:100). On the other hand, the positive dependence of the HCHO yield on humidity and the absence of OH scavenger is much less significant when the limonene:$O_3$ concentration ratio was high (1:2). There is no significant impact of OH scavenger on the HCHO yield found in this study consistent with findings in the experiments in Gong et al. (2018)."

**SECTION 3.1.2: This seems like a rather convoluted process to calculate the organic nitrate yield and I would suggest that, based on this, the stated uncertainty is rather low!**

Authors response:

The organic nitrate yield is calculated by performing a regression analysis between two cumulative quantities: the total amount of $RO_2$ that reacts with NO, and the total amount of organic nitrate present in the chamber which is calculated by the cumulative nitrogen production subtracting the concentrations of inorganic nitrogen species. The stated organic

nitrate yield (34±5%) here has a precision of about 15%, which is determined by the data points and their error bars of the two experiments. The size of the error bar is calculated based on the precision of the instruments with linear error propagation. The final precision (15%) is smaller than the precision of some of the measurements. This is because the precision of a cumulative quantity gradually increases when there are more data point available. On the other hand, the 1-σ accuracy of the nitrate yield is estimated to be about 30% at maximum, which is mostly attributed to the accuracy of the reaction rate constant $k_{RO_2+NO}$ (~30%) and the 1-σ accuracies of the HONO (10%) and $j_{HONO}$ (18%) measurements.

Statements are added to Line 444 – 447: "The precision (~15%) of $\Phi_{\text{limonene+OH}}$ is determined by the precision of the measurements with linear error propagation. The error of $\Phi_{\text{limonene+OH}}$ is estimated to be about 30%, which is mainly attributed to the accuracies of the reaction rate constants $k_{RO_2+NO}$ (~30%) and the measurements of HONO (10%) and $j_{HONO}$ (18%)."

**Line 679: This seems like the more likely explanation. You will be forming very different RO₂.**

Authors response:

We do not know the exact reason that causes the large difference in $k_{add}$. Although RO$_2$-limOH and RO$_2$-limO$_3$ are very different and have different chemistry, the difference in the fraction of RO$_2$-limOH and RO$_2$-limO$_3$ in the low-NO experiment and the ozonolysis experiment was too small to explain the large difference in $k_{add}$, as both experiments have about (40 – 60)% of RO$_2$-limOH and RO$_2$-limO$_3$. Therefore, temperature difference may be an important factor that causes the lower $k_{add}$ in the ozonolysis experiment. However, we do not want to make a solid conclusion on which factor is more likely. The sentence (Line 708 – 712) is rephrased to mention that temperature and the structure of the RO$_2$ could both contribute to the large difference in $k_{add}$: "The large difference in $k_{add}$ could be attributed to the different RO$_2$ species that are formed from the photooxidation reaction and the ozonolysis reaction. RO$_2$ formed from the photo-oxidation reaction have retain their 6-member ring moiety, whereas the majority of RO$_2$ formed from the ozonolysis reaction are acyclic. In addition, the low temperature during the ozonolysis experiment could slow down the additional loss pathway."

**Figure 9 and 10: Which experiment is which? Can these be labelled a and b.**

Authors response:

Thanks for the suggestion. Each subplot in Figure 9 and 10 is now labelled.

**Line 794: 'optimised'**

Authors response:

Thanks for the correction, it is corrected (Line 829).

**Responses to the comments from Referee 2:**

We thank the reviewer for the useful comments improving our manuscript.

Line 138: Is there an estimate for the "small" fraction of limonene-$RO_2$ that is converted and measured as $HO_2$ in the LIF detection cell during these experiments? Has this fraction been determined specifically for limonene-$RO_2$ and the NO concentrations used in detection cell or is it possible that this $RO_2$ interference is more significant than anticipated? If so, could this at least partially explain the discrepancies between measured and modeled $HO_2$ concentrations, especially during the ozonolysis experiment when measured $RO_2$ concentrations were highest?

**Authors response:**

A potential interference was only once explicitly tested, when the first experiments were performed. Results showed that the upper limit of an interference would be around 15%, but data were too noisy to derive an accurate number. We added in Line 144 to 146: "The upper limit of such an interference would be around 15% as indicated by characterization experiments, which unfortunately did not allow to determine an accurate number due to the limited precision of results."

Line 143: Are the $RO_2$ concentrations reported from all experiments derived from calibrations with methylperoxy radicals? If so, does this imply that the reported $RO_2$ concentrations, which are largely due to limonene-$RO_2$, represent a lower limit? Or have adjustments been made that take the $RO_x$-LIF system's reduced sensitivity to limonene-$RO_2$ into consideration?

**Authors response:**

$RO_2$ concentrations reported here are derived from the calibrations with methyl peroxy radicals. The $RO_x$-LIF measurement sensitivity of limonene-$RO_2$ relative to $CH_3O_2$ was determined in laboratory experiments to be 0.85±0.05. No correction is applied to account for the lower sensitivity, because this would require knowledge of the exact distribution of $RO_2$ radicals in the experiments. Corrections would be smaller than the discrepancies between modeled and measured $RO_2$ concentrations.

**line 207: Are the fluctuations in NO mixing ratios (and ultimately measured and modeled radical concentrations) during the low and medium NO experiments (Figures 3 and 4) caused by changes in HONO production from the chamber source that are driven by changes in solar**

**radiation? If so, these fluctuations may be easier for readers to interpret if measured or parameterized HONO mixing ratios or measurements of photolysis frequencies were shown.**

**Authors response:**

The fluctuation in NO mixing ratios is mainly driven by the fluctuation in the photolysis frequencies that are affected by cloud cover. In order to illustrate the effect, the photolysis frequency of HONO ($j_{HONO}$) is now added to the overview plots in photooxidation experiments (Figure 4, 5, 6, S3, S4, and S5).

**Figure S3: This figure is not discussed in the context of the low NO experiments. This is understandable since only a small portion of this experiment involves limonene oxidation, but since the figure is shown – are the observed RO$_2$ concentrations prior to the CH$_4$ addition likely due to the oxidation of some VOC produced in the chamber? It is interesting that, after the CH$_4$ injection, the measured RO$_2$ concentration increases as expected (at least relative to the established background), but the measurement/model agreement quickly reverses after limonene addition. Could this difference in measurement/model response to the different VOCs be related to the previously mentioned RO$_x$LIF sensitivities to CH$_3$O$_2$ and limonene-RO$_2$? Similarly, the model agrees with the HO$_2$ measurements during the CH$_4$ injection but underpredicts the measurements after the limonene injection. While these trends could again indicate a limonene-RO$_2$ interference in the HO$_2$ measurement, they could also support the later claims of missing RO$_2$ loss processes, whether isomerization or RO$_2$ + RO$_2$ recombination reactions, that are much faster for large complex monoterpene peroxy radicals (and produce HO$_2$), but do not occur for smaller RO$_2$ species like CH$_3$O$_2$. A short discussion on this particular experiment could be useful but is not absolutely necessary.**

**Author response:**

The observed RO$_2$ concentration before the injection of CH$_4$ is indeed likely due to the oxidation of unidentified background sources. The presence of these unidentified background sources is also seen in the background OH reactivity. Our model treats these unidentified background species as having the same chemical properties as CO to match the background OH reactivity. Therefore, RO$_2$ produced from the oxidation of the background source cannot be reproduced by our model calculations. The high measured HO$_2$ concentration right after the limonene injection would require that half of the RO$_2$ is detected in the HO$_2$ cell, which would be inconsistent with our characterization experiments, which determined an upper limit of the interference of 15%. Overall results from this experiment are consistent with results discussed for the experiment on 01 September 2012 in the main paper, so that we do not think that there is additional discussion needed.

**Line 533: "concentration" can be removed, or this sentence should be otherwise rephrased.**

**Author response:**

The sentence is corrected as suggested by removing "concentrations". (Line 561)

**Line 619: This sentence is a bit awkward. Perhaps "In the ozonolysis experiment, prior to the addition of CO as an OH scavenger (Fig. 8d) OH is only produced by the ozonolysis of limonene."**

**Author response:**

The sentence is changed as suggested by the reviewer. (Line 647 – 649)

**Line 659: Delete "-" after OH**

**Author response:**

The sentence is corrected as suggested. (Line 689)

**Figures 9, 12, and others in supplement: When data from multiple experiments are presented in one figure it would be useful to also label each panel (or group of panels) with "low NO" or "ozonolysis" instead of just the date. Figures 8 and S6 are good examples.**

**Authors response:**

Thanks for the suggestion. Figures 9 to 11 and S8 to S12 are now labeled with case name in each subplot. For Figure 12, experiment cases are now labeled separately (Medium NO: Figure 12; Low NO: Figure 13; Ozonolysis: Figure 14) similar to that of Figures S10 to S12. The Figures of the autooxidation mechanism of limonene-$RO_2$ are now Figure 15 and 16 (Line 934, 940). The labels in the text are also changed accordingly.

**Figures 9 and S8: The caption in Figure 9 suggests that $CH_3O_2$ is mainly produced from the oxidation of HCHO while the caption in Figure S8 suggests that $CH_3O_2$ is mainly produced from the oxidation of limonene.**

**Authors response:**

$CH_3O_2$ is mainly produced from the oxidation of HCHO that is produced from the oxidation of limonene in most of the experiments. In the low-NO experiment on 13 June 2015, $CH_3O_2$ was mainly produced from the oxidation of $CH_4$ that was added before the injection of limonene. The caption in Figure S8 is now changed to: "Methylperoxy radicals (CH3O2) are mainly produced from the oxidation of HCHO in most of the experiments or from the oxidation of $CH_4$ during the experiment on 13 June 2015" in Line 135 in the supplementary material.

**Lines 716, 720, 731, 1003: Some commas are unnecessary.**

**Authors response:**

The sentences are corrected. (Lines 750, 754, 765, 1035)

**Line 764/765: This sentence is a bit awkward. Consider "These reactions could involve an unknown reaction partner X, as used in Hofzumahaus et al. (2009), or could be unimolecular reactions." Also, this reference may be missing from the reference list.**

**Authors response:**

The sentence is changed as suggested and the missing reference is added. (Line 798 – 799)

**Line 893: One example instead of one examples.**

**Authors response:**

The sentence is corrected as suggested. (Line 934)

**Line 1018: Second "in the model" is unnecessary.**

**Authors response:**

The sentence is corrected as suggested. (Line 1050)

Reference:

Cox, R. A., Ammann, M., Crowley, J. N., Herrmann, H., Jenkin, M. E., McNeill, V. F., Mellouki, A., Troe, J., and Wallington, T. J.: Evaluated kinetic and photochemical data for atmospheric chemistry: Volume VII – Criegee intermediates, Atmos. Chem. Phys., 20, 13497–13519, https://doi.org/10.5194/acp-20-13497-2020, 2020.

Gong, Y., Chen, Z., and Li, H.: The oxidation regime and SOA composition in limonene ozonolysis: roles of different double bonds, radicals, and water, Atmos. Chem. Phys., 18, 15105–15123, https://doi.org/10.5194/acp-18-15105-2018, 2018.

Holland, F., Hessling, M. and Hofzumahaus, A.: In Situ Measurement of Tropospheric OH Radicals by Laser-Induced Fluorescence—A Description of the KFA Instrument, J. Atmos. Sci., 52, 3393 – 3401, https://doi.org/10.1175/1520-0469(1995)052%3C3393:ISMOTO%3E2.0.CO;2, 1995

Larsen, Bo. R., Di Bella, D., Glasius, M., Winterhalter, R., Jensen, N. R., and Hjorth, J.: Gas-Phase OH Oxidation of Monoterpenes: Gaseous and Particulate Products, J. Atmos. Chem., 38, 231–276, https://doi.org/10.1023/A:1006487530903, 2001.

Librando, V. and Tringali, G.: Atmospheric fate of OH initiated oxidation of terpenes. Reaction mechanism of α-pinene degradation and secondary organic aerosol formation, J. Environ. Manage., 75, 275–282, https://doi.org/10.1016/j.jenvman.2005.01.001, 2005.

Otkjær, R. V., Jakobsen, H. H., Tram, C. M., and Kjaergaard, H. G.: Calculated Hydrogen Shift Rate Constants in Substituted Alkyl Peroxy Radicals, J. Phys. Chem. A, 122, 8665–8673, https://doi.org/10.1021/acs.jpca.8b06223, 2018.

Praske, E., Otkjær, R. V., Crounse, J. D., Hethcox, J. C., Stoltz, B. M., Kjaergaard, H. G., and Wennberg, P. O.: Intramolecular Hydrogen Shift Chemistry of Hydroperoxy-Substituted Peroxy Radicals, J. Phys. Chem. A, 123, 590–600, https://doi.org/10.1021/acs.jpca.8b09745, 2019.

Vereecken, L., Novelli, A. and Taraborrelli, D.: Unimolecular decay strongly limits the atmospheric impact of Criegee intermediates, Phys. Chem. Chem. Phys., 19, 31599 – 31612, https://doi.org/10.1039/C7CP05541B, 2017